# IMPROVING OUT-OF-DISTRIBUTION GENERALIZATION WITH INDIRECTION REPRESENTATIONS

**Kha Pham** [1]**, Hung Le** [1]**, Man Ngo** [2]**, Truyen Tran** [1]

[1] Applied Artificial Intelligence Institute, Deakin University
[2] Faculty of Mathematics and Computer Science, VNUHCM-University of Science
[1]`{phti, thai.le, truyen.tran}@deakin.edu.au,`
[2]`nmman@hcmus.edu.vn,`

## ABSTRACT

We propose a generic module named Indirection Layer (InLay), which leverages indirection and data internal relationships to effectively construct symbolic indirect representations to improve out-of-distribution generalization capabilities of various neural architectures. InLay receives data input in the form of a sequence of objects, treats it as a complete weighted graph whose vertices are the objects and edge weights are scalars representing relationships between vertices. The input is first mapped via indirection to a symbolic graph with data-independent and trainable vertices. This symbolic graph is then propagated, resulting in new vertex features whose indirection will be used for prediction steps afterward. Theoretically, we show that the distances between indirection representations are bounded by the distances between corresponding graphs, implying that unseen samples with very different surface statistics can still be close in the representation space to the seen samples if they share similar internal relationships. We demonstrate that InLay is consistently effective in improving out-of-distribution generalization throughout a comprehensive suite of experiments, including IQ problems, distorted image classification, and few-shot domain adaptation NLP classification. We also conduct ablation studies to verify different design choices of InLay.

## 1 INTRODUCTION

There have been several evidences showing that deep learning models may fail drastically in out-of-distribution (OOD) testing circumstances (Geirhos et al., 2018; Keysers et al., 2020). One reason widely agreed upon is that neural networks tend to learn surface statistics of data (Lake et al., 2017) and thus can not generalize to new samples with different statistics. On the other hand, humans excel at generalizing, and it has been long believed that the ability to think in a symbolic way is the key for humans to quickly adapt to new situations (Mitchell, 2021). A powerful concept that can bridge concrete data and symbols is indirection, which binds two objects together and uses one to refer to the other. In computer science, indirection is widely used via pointer: data is bound to its memory address, and programs use the memory address to refer to that data.

The capacity to draw analogies is yet another trait that facilitates human generalization. Several cognitive science theories have been proposed to explain analogy, and the Structure-Mapping Theory (SMT) (Gentner, 1983) is one of the most successful among them. SMT argues that not object attributes but the relationships between them are transferred in an analogy. For example, the hydrogen atom is analogous to the solar system not because they share the same sizes or temperatures but because they both have entities revolving around a center due to the attractive force. This suggests that internal relationships of a situation contain essential information for generalization. In this paper, we propose a method that simultaneously leverages indirection and data internal relationships to construct *indirection representations*, which can be interpreted as symbolic representations that respect the similarities *between* internal relationships. For instance, two IQ problems with similar hidden rules (i.e., similar internal relationships) should have similar indirection representations, though they contain completely different shapes or images.

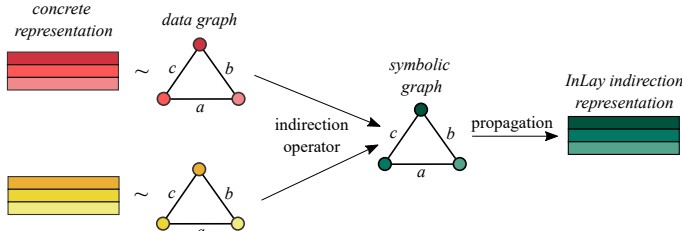

Figure 1: *Indirection Layer*. Concrete data representation is viewed as a complete graph with weighted edges. The indirection operator maps this graph to a symbolic graph with the same weight edges, however the vertices are fixed and trainable. This symbolic graph is propagated and the updated node features are indirection representations. Different concrete inputs may share the same indirection representations if their corresponding graphs have the same adjacency matrices. This illustrates the core idea of InLay: constructing indirection representations by transferring internal relationships through indirection.

To this end, we implement our method in the form of a generic module named Indirection Layer (InLay), which can construct indirection representations from either encoded or raw low-sensory data and can be equipped with various models to improve their OOD generalization capabilities. InLay receives a sequence of objects as input and produces a sequence with the same length including associated indirection representations. The input sequence is viewed as a complete weighted graph where each edge weight represents the relationship between two corresponding objects, and thus the adjacency matrix of this graph captures the internal relationships of the input. The core operation of InLay consists of two steps: indirection and graph propagation (see Fig. 1 for illustration). The input is first processed through indirection to transfer all edge weights to another symbolic graph whose vertices are data-independent and trainable. This symbolic graph is then propagated, resulting in updated vertex features as the indirection representations of the input. These indirection representations are used as new representations for prediction steps afterward.

We show both theoretically and empirically that InLay can help to improve OOD generalization. Theoretically, we show that InLay indirection preserves internal structures of graphs, and the distances between indirection representations are bounded by the cut distances between corresponding graphs. Thanks to these theoretical properties, the indirection representation of a new data instance can be located near a seen one if they share similar internal relationships (although the surface features may be entirely different), thus the two instances have a higher chance of being interpreted similarly. Empirically, we show that InLay consistently helps different models to improve their OOD generalization capabilities in a comprehensive suite of experiments involving numerous datasets and OOD scenarios, including IQ problems with unseen objects and unseen rules, distorted image classification, and few-shot domain adaptation NLP classification. We also conduct ablation experiments to study the necessity of different design choices in InLay and provide practical analysis on the success of InLay.

## 2 METHOD

We introduce our main contribution, namely the Indirection Layer (InLay). InLay takes a sequence of objects as input and transforms the sequence into a new indirect graph-structured representation. Concretely, let $X = (x_1, x_2, \ldots, x_k)^\top \in \mathbb{R}^{k \times n}$ be the input sequence for InLay, where $k$ is the number of objects and each $x_i \in \mathbb{R}^n$ represents an object. For example, an object may be either an image in IQ problems, or a patch of image in image classification task, or a paragraph in few-shot NLP classification task (see Section 4). To better exploit data internal relationships, we treat each input sequence as a directed complete weighted graph (with no self-loop) whose vertices represent the objects and edges represent relationships as scalars in $[-1, 1]$. Specifically, for each sequence $X$, we denote $G_X$ as its corresponding graph. We define $\mathcal{G}_k$ to be the space of all directed weighted complete graphs $G$ with $k$ vertices and edge weights in $[-1, 1]$. From now on, we will only write $G$ instead of $G_X$ when it is not necessary to specify $X$, and we denote $A_G$ as the adjacency matrix of $G$. This adjacency matrix captures the internal relationships of the corresponding data sequence.

*Remark* 2.1. (*Canonical indexing assumption*) As the set of graph vertices may permute, a graph $G$ with $k$ vertices may not have an unique adjacency matrix. To assure the well-definedness of $A_G$, we assume that (when computing the adjacency matrix) the $i$-th vertex represents the $i$-th element of the input sequence. We show in Appendix C that the indirection representations are still maintained if the canonical indexing assumption is not obeyed.

We aim to learn suitable representations for the graph such that the internal relationships of the input sequence can be transferable to novel settings. To this end, we contribute the Indirection Layer (InLay), which leverages indirection and data internal relationships to construct indirection representations. InLay is a generic and flexible module that can be equipped into different models to construct indirection representations from either encoded data or raw low-sensory data (e.g, for the case of Vision Transformer; see Section 4.2) in two steps: indirection and graph propagation.

**Indirection**

For each $X$, the adjacency matrix $A_{G_X} \in \mathbb{R}^{k \times k}$ of $G_X$ represents the internal relationships between objects in $X$. Each component $a_{ij}^X$ of $A_{G_X}$ is computed as $a_{ij}^X = \tanh\left(\frac{Q^\top x_i \cdot K^\top x_j}{\sqrt{4n}}\right)$ if $i \neq j$ and $a_{ij}^X = 0$ if $i = j$, where $\cdot$ is the inner product and $Q, K \in \mathbb{R}^{n \times 4n}$ are trainable weights that project $x_i$ and $x_j$ onto a higher dimensional space so that a linear kernel may represent the relationship between $x_i$ and $x_j$. The choice of tanh as a non-linear transformation is important: it maps the dot products to $[-1, 1]$, allowing InLay to possess nice theoretical properties regarding boundedness of distances (see Section 3); and, tanh allows negative similarities between objects, which may help to represent opposite relations, e.g., translations to the left and to the right. See Section 4.1.1 and Appendix G for experimental details.

In indirection, each object is bound to a symbol. We denote by $V^{\text{ind}} = \left(v_1^{\text{ind}}, v_2^{\text{ind}}, \ldots, v_k^{\text{ind}}\right)^\top \in \mathbb{R}^{k \times n}$ to be the set of symbols where each $v_i^{\text{ind}} \in \mathbb{R}^n$ is data-independent and trainable. Let $\mathcal{G}_k^{\text{ind}}$ be the subset of $\mathcal{G}_k$ that consists of all graphs whose set of vertices is $V^{\text{ind}}$ (i.e., each vertex represents some $v_i^{\text{ind}}$ and no two vertices represent the same $v_i^{\text{ind}}$). The space $\mathcal{G}_k^{\text{ind}}$ can be interpreted as the space of symbolic graphs with fixed vertices. We define the indirection operator $\mathcal{I}$ as follows.

**Definition 2.2.** Given an input sequence $X = (x_1, x_2, \ldots, x_k)$ and its corresponding graph $G_X \in \mathcal{G}_k$, the *indirection operator* $\mathcal{I}$ is a mapping from $\mathcal{G}_k$ to $\mathcal{G}_k^{\text{ind}}$ that maps $G_X$ to $\mathcal{I}(G_X)$ so that $A_{G_X} = A_{\mathcal{I}(G_X)}$ and the $i$-th vertex of $\mathcal{I}(G_X)$ represents $v_i^{\text{ind}}$.

*Remark* 2.3. Definition 2.2 is introduced in the case when the canonical indexing assumption (see Remark 2.1) is obeyed. The vertex order emerges when computing the adjacency matrix. A more general definition is given in Appendix C.

The indirection operator $\mathcal{I}$ maps each object $x_i$ to its associated symbol $v_i^{\text{ind}}$ while assuming the pairwise relationship between $v_i^{\text{ind}}$ and $v_j^{\text{ind}}$ is the same as one between $x_i$ and $x_j$ (see Fig. 1). That is, $\mathcal{I}$ ignores the concrete features of objects but still maintains the relationships between them.

**Graph propagation**

After indirection, each data graph $G$ is mapped to a symbolic graph $\mathcal{I}(G)$. This operation can be interpreted as follows: at first, edge weights of $\mathcal{I}(G)$ are unspecified; then the indirection operator $\mathcal{I}$ assigns edge weights from the data to $\mathcal{I}(G)$. Once receiving this information from data, $\mathcal{I}(G)$ is propagated and the updated vertex features are indirection representations of the input sequence. Formally, for an input sequence $X$, if we denote $r_X$ to be the indirection representations of $X$, then $r_X = A_{G_X} V^{\text{ind}}$. This symbolic $r_X$ is used as a new representation for $X$ for prediction steps afterward.

To summarize, for each input sequence $X$, InLay constructs associated indirection representation $r_X$:

$$r_X = \tanh\left(\frac{XQ(XK)^\top}{\sqrt{4n}}\right) V^{\text{ind}}. \tag{1}$$

This equation is closely related to self-attention, except for three points: 1. data are projected onto a higher dimensional space by matrix multiplying with $Q$ and $K$; 2. the softmax operator is replaced by tanh; and most importantly, 3. the value $V^{\text{ind}}$ is **not** computed based on the data $X$. While the first two differences empirically enhance InLay's performances (see Section 4.4), the third one stands for the core idea of indirection in InLay. An ablation study on $V^{\text{ind}}$ will also be conducted in Section 4.4 to demonstrate the role of each element.

Initialization of $V^{\text{ind}}$ may greatly affect the overall performance. To reduce this effect, we replace $V^{\text{ind}}$ in Eq. (1) by $\psi(V^{\text{ind}})$, where $\psi : \mathbb{R}^n \to \mathbb{R}^n$ is a trainable 2-layer neural network applied to rows of $V^{\text{ind}}$. We also use multi-heads to compute the adjacency matrix so that local information of feature vectors is better utilized. The number of heads is tuned for each specific task.

## 3 THEORETICAL ANALYSIS

### 3.1 BOUNDEDNESS WITH RESPECT TO THE CUT DISTANCE

Graph spectrum and Laplacian are important graph characteristics that can be computed entirely by graph adjacency matrices. From Definition 2.2, it follows that the indirection operator $\mathcal{I}$ preserves graph spectrum and Laplacian, which means $\mathcal{I}$ preserves graph internal structure. To some extent, this agrees with the Structure-Mapping Theory (Gentner, 1983), which states that not the attributes but the internal relationships are transferred in an analogy. In other words, learning internal relationships may already be enough to capture the essence of a situation. Further details for the Structure-Mapping Theory will be given in Section 5.

Next, we investigate how distances between graphs may constrain distances between indirection representations. Before defining graph distance, we define isomorphism between graphs in $\mathcal{G}_k$.

**Definition 3.1.** Given two graphs $G = (V, E) \in \mathcal{G}_k$ and $G' = (V', E') \in \mathcal{G}_k$ with associated adjacency matrices $A_G = (a_{ij})_{i,j=\overline{1,k}}$ and $A_{G'} = (a'_{ij})_{i,j=\overline{1,k}}$. We say $G$ and $G'$ are *isomorphic*, denote by $G \cong G'$, if there exists a bijection $\phi : V \to V'$ so that $a_{ij} = a'_{\phi(i)\phi(j)}$ for every $i, j \in V$.

Two isomorphic graphs can be interpreted as being identical up to isomorphism, and thus a graph distance defined on $\mathcal{G}_k$ should respect this property, i.e., the distance between two isomorphic graphs is 0. One such distance is the cut distance $\hat{\delta}_\square$ (Borgs et al., 2008), which is a useful tool to compare similarities between structures (Liu et al., 2018), and also for studying the convergence of sequence of graphs (Borgs et al., 2008). A formal definition for $\hat{\delta}_\square$ is given in Appendix A.

It follows from the definition of $\hat{\delta}_\square$ that $\hat{\delta}_\square(G, G') = 0$ if and only if $G$ is isomorphic with $G'$. Moreover, if $G_1 \cong G_2$ then $\hat{\delta}_\square(G_1, G') = \hat{\delta}_\square(G_2, G')$ for any $G'$. Since $G \cong \mathcal{I}(G)$, the indirection operator $\mathcal{I}$ preserves $\hat{\delta}_\square$ distance, i.e., $\hat{\delta}_\square(G, G') = \hat{\delta}_\square(\mathcal{I}(G), \mathcal{I}(G'))$ for every $G, G' \in \mathcal{G}_k$.

We have shown that the indirection operator $\mathcal{I}$ admits invariant properties with respect to the graph spectrum, Laplacian and the cut graph distance $\hat{\delta}_\square$. The following result shows that the distances between indirection representations are bounded by cut distances between corresponding graphs. For each $G \in \mathcal{G}_k$, we denote $r_G = A_G V^{ind}$ to be its associated indirection representation.

**Theorem 3.2.** *For any two graphs $G \in \mathcal{G}_k$ and $G' \in \mathcal{G}_k$, the following inequality holds:*

$$\|r_G - r_{G'}\|_\infty \leq k \left( 2 + k^2 \hat{\delta}_\square(G, G') \right) \|V^{ind}\|_\infty, \tag{2}$$

*where $\|.\|_\infty$ is the matrix infinity norm (see Definition A.3 in Appendix A).*

*Proof.* See Appendix B. $\qquad\square$

Note that even when $\hat{\delta}_\square(G, G') = 0$, $r_G$ may still be different from $r_{G'}$. This is because by design, InLay also takes into account the ordinal information of input sequence, which may be important in some specific use cases, e.g., when the input is sequence of image patches. Theorem 3.2 shows that if $G$ and $G'$ are close, their indirection representations will not be far away from each other as well. This is an important property since the original vertex representations of $G$ and $G'$ may be arbitrarily far though $G$ and $G'$ are isomorphic, e.g., two IQ problems with the same hidden rules but different images may be represented very differently. Theorem 3.2 also shows the necessity of training $V^{ind}$ to obtain appropriate $\|V^{ind}\|_\infty$: if $\|V^{ind}\|_\infty$ is too large, the bound in Ineq. (2) is loose; conversely, if $\|V^{ind}\|_\infty$ is too small, the bound may be too strict so that indirection representations are not well separated enough. An empirical ablation study on $V^{ind}$ will be given in Section 4.4.

### 3.2 CONNECTION BETWEEN INLAY AND STRUCTURAL ANALOGY

Current machine learning methods follow the manifold hypothesis and tend to interpolate on the learned manifold during testing. This ability of interpolation is usually referred as making value analogies, i.e., making analogies between data features. However, value analogy may not be enough in more extreme generalization cases when surface statistics of testing samples vastly differ from that of training data. Structural analogy is believed to be necessary for ML models to reach higher levels of generalization (Chollet, 2021). By making structural analogy, concrete information is partly ignored while structural information is compared, e.g., two IQ problems with the same hidden rules

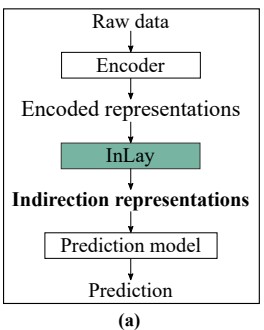 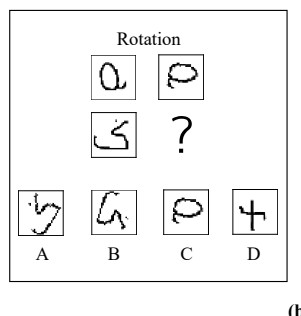 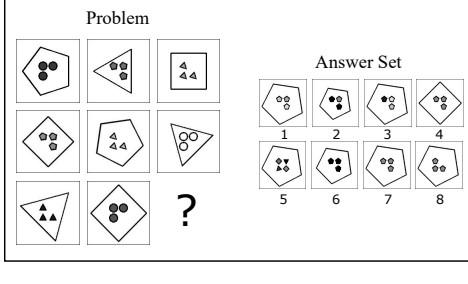

     (a)                             (b)

Figure 2: **(a)** Overall architecture of prediction model when equipped with InLay. **(b)** *Left:* A problem in FINE dataset with images from the Omniglot dataset and hidden rule is 90-degree rotation. *Right:* A Raven's Progressive Matrix problem from RAVEN dataset. Images adapted from original papers.

are structurally analogous even though the data (e.g., images) are entirely different between the problems. In InLay, the structural information is maintained in the form of adjacency matrices, which are computed based on data features. To some extent, InLay can be interpreted as a hybrid method of value analogy and structural analogy.

When it comes to structural analogy, one might need a metric to measure the similarities between structures. Among different metrics, the cut distance $\hat{\delta}_{\square}$ is one of the few methods able to compare directed weighted graphs (Tantardini et al., 2019). For instance, a recent work by Liu et al. (2018) leverages the cut distance to compare complex networks, including artificial networks and real networks of chemical molecules. Theorem 3.2 draws a connection between InLay and the cut distance by showing that the distances between indirection representations are bounded by corresponding cut distances, and thus emphasizes the structural inductive bias InLay brings into deep learning models.

## 4 EXPERIMENTS

In this section, we conduct several experiments with different scenarios of OOD generalization to show that InLay can adapt to various models and improve their performances on various datasets. The OOD testing scenarios include IQ problems with unseen images and unseen rules, distorted image classification, and domain adaptation on few-shot NLP classification, all of which require the ability to understand the problem in a systematic and symbolic way in order to generalize on new OOD circumstances. Throughout these experiments, we show that InLay consistently helps models to perform better. We also provide an ablation study on the necessities of different design choices in InLay, as well as a practical analysis of the success of InLay.

InLay constructs indirection representations from either encoded or raw data. If the prediction model is equipped with an encoder, InLay will sit between the encoder and prediction to transform encoded representations to indirection representations (see Fig. 2a for an illustration). When there is no encoder, e.g., when the prediction model is Vision Transformer, InLay directly constructs indirection representations from raw data. To be fair when comparing, models with or without InLay are all trained with the same training settings, including batch size, learning rate, number of training iterations, optimizer, etc., and we only report test results after the last iteration. Average results are reported in the main text; full results with standard deviation are given in Appendix D. More training details are also given in Appendix K.

In practice, we optionally use context normalization (Webb et al., 2020a) to further improve InLay. If context normalization is applied in InLay, there will be two such layers: one to normalize the original representation $X$, and one to normalize the symbolic representation $r_X$.

### 4.1 OUT-OF-DISTRIBUTION IQ PROBLEMS

IQ problems are powerful testbeds for OOD generalization capability of deep learning models. Despite their simple appearances, IQ problems are challenging in the sense that they require models to understand the hidden rules instead of just surface features to solve new problems with unseen objects or even unseen (but related) rules. There have been evidences showing that current deep learning models may fail when facing problems with unseen objects (Webb et al., 2020b). In this experiment, we show that models coupled with InLay achieve better performances on two IQ datasets: FINE (Pham et al., 2022) and RAVEN (Zhang et al., 2019). Examples are given in Fig. 2b.

| Train set | CIFAR100 | | | | MNIST | | | |
| Test set | Omniglot | | | | CIFAR100 | | | |
| | Trans. | Rot. | Shear | Scale | Trans. | Rot. | Shear | Scale |
|---|---|---|---|---|---|---|---|---|
| NTM | 25.5/**56.9** | 27.1/**31.4** | 29.8/**58.5** | 30.3/**61.0** | 35.2/**69.2** | 30.0/**34.5** | 29.6/**56.0** | 31.9/**48.7** |
| PrediNet | 26.7/**28.5** | 25.5/**28.3** | 25.6/**29.0** | 26.4/**29.9** | 27.5/**31.6** | 26.8/**27.8** | 26.4/**28.4** | 33.6/**35.9** |
| RelationNet | 25.4/**39.3** | 25.0/**34.0** | 25.1/**39.8** | 24.7/**50.2** | 25.6/**45.5** | 25.6/**35.2** | 25.0/**43.4** | 26.8/**44.8** |
| Transformer | 26.9/**63.3** | 27.2/**40.4** | 28.3/**65.3** | 30.6/**62.0** | 37.8/**63.9** | 32.0/**36.1** | 30.2/**53.5** | 29.2/**48.1** |

Table 1: Average test accuracy (%) without/with InLay on FINE dataset.

| Train configuration | Up-Down | 3x3Grid | Out-InGrid | Average |
| Test configuration | Left-Right | 2x2Grid | Out-InCenter | |
|---|---|---|---|---|
| LSTM | 28.8/**43.9** | 19.5/**25.1** | 42.1/**48.6** | 30.1/**39.2** |
| Transformer | 15.2/**56.7** | 13.6/**26.0** | 16.6/**44.7** | 15.1/**42.5** |
| RelationNet | 12.7/**58.5** | 12.4/**28.9** | 12.3/**51.7** | 12.5/**46.4** |
| PrediNet | 13.7/**16.5** | 13.6/**14.3** | 14.2/**15.9** | 13.8/**15.6** |

Table 2: Average test accuracy (%) without/with InLay on RAVEN dataset.

### 4.1.1 FINE DATASET

FINE dataset consists of IQ problems with geometric transformations as hidden rules. To succeed in this dataset, models should treat objects as symbols and learn the relationship between these symbols. Here we consider one of the most challenging OOD scenarios: test problems include unseen objects and unseen rules. To be specific, images in train and test problems come from different datasets (train on CIFAR100 (Krizhevsky, 2009) - test on Omniglot (Lake et al., 2015) or train on MNIST (LeCun et al., 2010) - test on CIFAR100), so the models need to understand the hidden rules instead of image features to solve test problems. Rules in test problems are also unseen during training; for instance, we train on IQ problems with rotation angle less than $180°$ and test on ones with rotation angles more than $180°$; similarly, for translation, we train on problems with translation to the left and test on ones with translation to the right. More details of other transformations will be given in Appendix. Models to be considered include the Neural Turing Machine (Graves et al., 2014), PrediNet (Shanahan et al., 2020), Relation Network (Sung et al., 2018), and Transformer (Vaswani et al., 2017), all of which have been shown to be effective on different relational reasoning tasks. All models, with or without InLay, are equipped with context normalization. Results are reported in Table 1.

Overall, models perform better when equipped with InLay, even in very extreme cases when models are trained on grayscale MNIST images and tested on RGB CIFAR100 images. This shows a clear advantage of InLay: indirection maps concrete features to symbolic space spanned by $V^{\text{ind}}$, thus helping models to be less dependent on concrete data. It can also be observed that PrediNet equipped with InLay improves less than other models. This is because PrediNet also constructs new symbolic data representations, and thus contains the inductive bias of symbolic representations itself. Again, this emphasizes the necessity of including symbolic inductive bias to improve OOD generalization.

### 4.1.2 RAVEN DATASET

RAVEN dataset (Zhang et al., 2019) is inspired by Raven's Progressive Matrices, which are challenging IQ tests for humans. RAVEN IQ problems are complex in the sense that a single problem may consist of different shapes, each of which follows a different rule. Inspired by the original paper, we conduct experiments to test models' capabilities to generalize on problems of unseen configurations. Specifically, we train and test on problems of different but related configurations. For example, Up-Down problems are related to Left-Right ones in the sense that Left-Right configuration can be viewed as a 90-degree "rotation" of Up-Down. In this experiment, the three train-test configuration pairs are UpDown-LeftRight, 3x3Grid-2x2Grid and OutInGrid-OutInCenter. We also apply context normalization in all models. We further apply Dynamic Residual Tree as proposed in the original paper to capture the inherent structure of Raven's Progressive Matrices. Results are reported in Table 2. A similar pattern can be observed: models equipped with InLay tend to perform better. In average, InLay helps improve LSTM by $9.1\%$, Transformer by $27.4\%$, RelationNet by $33.9\%$, and PrediNet by $1.8\%$. This can be well explained by Theorem 3.2 and Theorem B.1 in the theoretical analysis: although the test configuration is unseen, it is still related to the train configuration and thus InLay representations for test problems may still be close to ones of train problems. This helps models to have a better chance to draw analogies between observed and unobserved configurations.

| Dataset | No data augmentation | | | | With data augmentation | | | |
|---|---|---|---|---|---|---|---|---|
| | Rot90 | Grayscale | Jitter | *Avg.* | Rot90 | Grayscale | Jitter | *Avg.* |
| SVHN | 11.8/**12.5** | 86.9/**91.8** | 87.2/**91.6** | 62.0/**65.3** | 17.3/**19.1** | 92.9/**94.7** | 91.8/**93.4** | 67.3/**69.1** |
| CIFAR10 | 25.7/**27.9** | 48.3/**62.7** | 46.6/**59.7** | 40.2/**50.1** | 24.2/**27.3** | 54.2/**69.2** | 50.2/**68.4** | 42.2/**55.0** |
| CIFAR100 | 13.4/**15.0** | 16.4/**23.9** | 18.1/**25.0** | 16.0/**21.3** | 11.1/**14.0** | 18.5/**27.8** | 19.4/**28.7** | 16.3/**23.5** |

Table 3: Average test accuracy (%) without/with InLay on OOD classification task with ViT.

| | ProtoNet | SNAIL | GNN | MTB |
|---|---|---|---|---|
| 5-way-1-shot | 64.7/**65.4** | 38.6/**60.6** | 36.7/**63.1** | 66.1/**68.9** |
| 10-way-1-shot | **49.1**/41.7 | 17.3/**34.9** | 27.1/**43.3** | 52.9/**54.1** |
| Average | **56.9**/53.6 | 28.0/**47.8** | 31.9/**53.2** | 59.5/**61.5** |

Table 4: Average validation accuracy (%) without/with InLay on FewRel 2.0.

## 4.2 OUT-OF-DISTRIBUTION CLASSIFICATION

Humans can consistently recognize objects in different positions, angles, or colors. Current deep learning models may not. Geirhos et al. (2018) show that when test images are injected with different kinds of distortions other than ones in training, deep neural networks may fail drastically on image classification tasks. We take inspiration from that result and conduct similar experiments to test whether InLay can help models improve their performances on OOD image classification tasks. We use a vanilla 6-layer Vision Transformer (ViT) (Dosovitskiy et al., 2020) as the base model and test it, with or without InLay, on different datasets, namely the SVHN (Netzer et al., 2011) and CIFAR10&100 (Krizhevsky, 2009). The models are trained on original images in two cases: with and without data augmentation, and tested on images with various distortions, including image transformation (90-degree rotation) and color transformations (color jitter, grayscale). In the case of data augmentation, we use all other distortions for augmentation except one used for testing. If InLay is equipped, we divide each $32 \times 32$ image into overlapping patches of size $8 \times 8$ and stride 4. These patches are vertices of the graph that represents the current image. The patch indirection representations are reassembled to form a new image of the same size as the original one. This new image is then fed into ViT. Context normalization is not used in this task.

Results are shown in Table 3. As expected, ViT performs poorly when the images are distorted. In average, InLay helps to improve its performance by 3.3% on SVHN, 0.9% on CIFAR10, and 5.3% on CIFAR100 when there is no data augmentation and 1.8% on SVHN, 12.8% on CIFAR10, and 7.2% on CIFAR100 when data augmentation is included. We can also observe that 1. ViT with data augmentation but without InLay is still mostly worse than ViT with InLay but without data augmentation; and 2. InLay helps improve ViT in both cases of with and without data augmentation. Note that it should not be interpreted that adding InLay to ViT is equivalent to adding a Transformer layer. Empirically, performances of 7-layer ViT only slightly differ from 6-layer ViT (see Appendix E), while it is clear that adding InLay may boost performances significantly.

## 4.3 FEW-SHOT NLP DOMAIN ADAPTATION

Humans can handle NLP classification tasks given a small number of examples. While humans can quickly adapt to such new scenarios, deep language models may not. Gao et al. (2019) proposed the FewRel 2.0 dataset that consists of few-shot NLP classification tasks, where the domains of train and test tasks vastly differ. Specifically, the training texts are taken from the Wikipedia corpus, while texts for testing originate from the PubMed and UMLS databases that contain large amounts of biomedical literature and sciences. This creates a big obstacle for few-shot language models to adapt to: their performances drop drastically as reported in the original paper.

Inspired by this result, we conduct an experiment on FewRel 2.0 dataset to show that InLay works well for language models. Different few-shot models, including Prototypical Network (Snell et al., 2017), SNAIL (Mishra et al., 2017), Graph Neural Network (Garcia and Bruna, 2017), and MTB (Soares et al., 2019), are trained with BERT encoder (Devlin et al., 2018) on 5-way-1-shot and 10-way-1-shot tasks. All models are equipped with context normalization. Since the test set is not provided for the public, we only report the test results on validation set, which shares the same domain as the test set. Results are shown in Table 4. Except ProtoNet, InLay helps other models improve: 19.8% for SNAIL, 21.3% for GNN, and 2.0% for MTB in average. The case of ProtoNet can be explained as follows: ProtoNet depends the distances between data instances, which is similar to the spirit of our InLay. Because ProtoNet already has this inductive bias, InLay can not help to improve it.

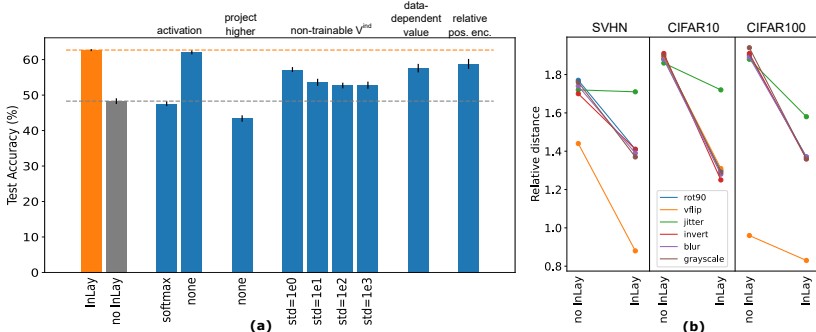

Figure 3: **(a)** *Ablation study.* Test accuracies of ViT equipped with InLay on grayscale images when a design choice of InLay is replaced or removed. **(b)** Relative distances between original and distorted representations with and without InLay in different testing cases.

## 4.4 ABLATION AND ANALYSIS

### 4.4.1 ABLATION ON INLAY DESIGN CHOICES

We conduct ablation experiments to study the necessities of different design choices of InLay. All experiments are conducted on OOD classification task (see Section 4.2) with ViT and grayscale testing images. In each experiment, we modify one design choice and keep others fixed. We consider three main design choices: activation function to compute adjacency matrices, projection on higher space to compute dot products, and trainability and data-independence of $V^{\text{ind}}$. We also consider the case when the indirection representations are treated as relative positional encoding to be added to the original input. Results are reported in Fig. 3a.

In the ablation for activation function, replacing tanh with softmax significantly decreases the performance, while the result when no activation is applied is only slightly lower. This is because softmax does not allow negative values; moreover, it imposes the constraint of summing-to-one on edges of graph, which is unnecessary in the theoretical analysis. On the other side, projecting data onto higher dimensional spaces also plays a vital role in InLay as it helps linearize the relations between objects so that dot products may manage to represent those relations, and not doing so may lead to a drastic drop in performance. Maintaining a trainable set of symbols $V^{\text{ind}}$ is beneficial for InLay, and the performances with randomly sampled $V^{\text{ind}}$ from Gaussian tend to decrease when the standard deviations of the Gaussians increase. This can be explained by Theorem 3.2: increase of standard deviations leads to bigger $\|V^{\text{ind}}\|_\infty$, which loosens the bound in Ineq. (2). Keeping $V^{\text{ind}}$ data-independent is also important, and treating the indirection representations as relative positional encoding is not efficient.

### 4.4.2 FURTHER PRACTICAL ANALYSIS

Besides theoretical analysis in Section 3, we further provide practical evidence showing why InLay may help models to generalize better. We again use the OOD classification tasks as a testbed. In short, we would like to show that InLay reduces the distance between an image and its distorted version, thus models may recognize the similarity between the two images more easily. Using the absolute distance may not be a fair metric since scaling two vectors by the same factor may already reduce the distance between them. Instead, we compute the relative distances between vectors, i.e. the relative distance between $u$ and $v$ is $2 \times \frac{\|u-v\|_\infty}{\|u\|_\infty + \|v\|_\infty}$. Relative distances between images and their distorted versions are computed in two cases: with InLay and without InLay. Results are shown in Fig. 3b, and it is clear that the relative distances in InLay case are lower than those without InLay. This can be partly explained by Theorem 3.2: the original image corresponds with $G$, and the distorted image corresponds with $G'$. Since the distorted image is closely related to the original one, the distance between their corresponding graphs is small, and thus the distance between their indirection representations $r_G$ and $r_{G'}$ also tends to be small according to Ineq. (2).

## 5 RELATED WORK

Systematic generalization has attracted attention recently in the deep neural networks community. One approach is to train a mixture of experts as functional modules, and these experts either compete (Parascandolo et al., 2018) or are composed by attention mechanism (Rahaman et al., 2021) to solve a task. Fedus et al. (2021) proposed the Switch Transformer to simplify routing algorithms in mixture-of-experts models to reduce communication and computational costs. Another approach is

to design architectures that mimic human's ability to think and reason sequentially. One well-known early model following this approach is the Module Network (Andreas et al., 2016) which attacks the image-QA tasks by parsing the query into sequential sub-queries, each of which is solved by a module in the form of neural networks. The MAC recurrent network (Hudson and Manning, 2018) and the Neural State Machine (Hudson and Manning, 2019) follow a similar idea, however, in MAC the query is explicitly and expressively decomposed by a sequence of RNN-type MAC cells, while Neural State Machine relies on probabilistic graphs representing underlying semantics to reason sequentially. Recently, Wei et al. (2022) proposed the idea of chain of thought to improve the ability of large language models to perform complex reasoning. Our InLay also follows the idea of injecting symbolic inductive bias, however, we focus on representations instead of functional modules. Models equipped with InLay can be interpreted as 2-step reasoning: the low-level sensory data is first represented symbolically by InLay, then processed by the following models.

Indirection is one of the most useful ideas that has been long applied in different areas of computer science. One of the most illustrative examples for indirection is the concept of pointer. Recently, there have been works leveraging indirection to improve generalization capabilities of deep learning models. ESBN (Webb et al., 2020b) uses an RNN controller to sequentially produce a symbolic key for each object and reasons on keys only. The keys in ESBN are computed based on the controller and similarities between objects, which is similar to our InLay; however, the keys are produced sequentially, which may increase the computational cost. Recently, Pham et al. (2022) proposed FINE, which is a fast-weight approach that utilizes indirection on functional spaces and has achieved promising performances on different OOD testing scenarios of IQ problems.

The idea of transferring relationships in InLay is inspired from the Structure Mapping Theory (SMT) (Gentner, 1983), which is a revolutionary theory of analogy in cognitive science. Analogy is a vital concept to explain human cognition, and it has been long argued that analogy-making underlies humans' ability to flexibly adapt to new situations (Gentner et al., 2001). Before SMT, it had been assumed that in a strong analogy, the base and the target should share several attributes in common (Tversky, 1977). SMT, in contrast, argues that not attributes but the relationships between objects are transferred in an analogy; in other words, the essence of a situation lies in internal relationships instead of concrete attributes. From this point of view, the theoretical results in Section 3 can be interpreted as a justification for SMT in the case of InLay: transferring relationships only does not lose significant information such as graph characteristics and graph topology.

It is also worth noting that from few-shot learning perspective, our InLay can be categorized as fast-weight (Malsburg, 1994) embedding learning model, in which the attentional weight $A_{G_X}$ is computed on-the-fly and the indirection representation $r_X$ is computed accordingly.

The trainable set of symbols $V^{\text{ind}}$ can be interpreted as positional encodings, and the output $r_X$ of InLay is a relative positional encoding regarding the input $X$. The idea of relative positional encoding (Shaw et al., 2018), as a replacement for the absolute positional encoding in Transformer, has been widely investigated and several variants have been proposed (Dai et al., 2019; Huang et al., 2018). It is worth noting that the relative positional encoding is added to the original input, while $r_X$ in InLay plays the role of *the* input for the following model. InLay thus should not be considered as a variant of a Transformer layer or relative positional encoding; instead, its design highlights the idea of indirection and symbolic representations.

## 6 CONCLUSION

In this paper, we propose InLay as a separate module that can be plugged into different models to improve OOD generalization. InLay leverages the idea of indirection to redirect data representation based on a trainable set of symbols. Viewing each data point as a complete weighted graph, we prove theoretically that InLay preserves graph internal structure and graph topology, and the distances between refined representations are bounded by the distances between corresponding graphs. We show the effectiveness of InLay through a comprehensive suite of experiments, including different OOD testing scenarios on IQ problems, distorted image classification, and few-shot NLP domain adaptation classification tasks. We also conduct ablation experiments to study necessities of different design choices in InLay, as well as further practical analysis on the success of InLay.

InLay opens up several future directions. From the theoretical side, it is worth investigating how the manifold containing original data representations is transformed during InLay, and why this manifold transformation can help generalization. From the practical view, stacking multiple InLay's to form a hierarchical indirection network is a promising idea.

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

# APPENDIX

## A  DEFINITION OF $\hat{\delta}_\square$ AND $\|.\|_\infty$

We first define the cut distance $d_\square$ between graphs with the same set of vertices.

**Definition A.1.** (Borgs et al., 2008) Given two graphs $G = (V, E) \in \mathcal{G}_k$ and $G' = (V, E') \in \mathcal{G}_k$ with associated adjacency matrices $A_G = (a_{ij})_{i,j=\overline{1,k}}$ and $A_{G'} = (a'_{ij})_{i,j=\overline{1,k}}$. The *cut distance* $d_\square$ between $G$ and $G'$ is

$$d_\square(G, G') = \max_{S,T \subset V} \frac{1}{k^2} |e_G(S, T) - e_{G'}(S, T)|, \tag{3}$$

where

$$e_G(S, T) = \sum_{i \in S, j \in T} a_{ij} \quad \text{and} \quad e_{G'}(S, T) = \sum_{i \in S, j \in T} a'_{ij}.$$

The distance $d_\square$ can also be interpreted as the distance between the internal relationships (i.e., the adjacency matrices) of two graphs. However, one drawback of $d_\square$ is that it is not invariant under isomorphism. The generalized cut distance $\hat{\delta}_\square$ is proposed to overcome this drawback.

**Definition A.2.** (Borgs et al., 2008) Given two graphs $G = (V, E) \in \mathcal{G}_k$ and $G' = (V, E') \in \mathcal{G}_k$ with associated adjacency matrices $A_G = (a_{ij})_{i,j=\overline{1,k}}$ and $A_{G'} = (a'_{ij})_{i,j=\overline{1,k}}$. The *generalized cut distance* $\hat{\delta}_\square$ between $G$ and $G'$ is computed as $\hat{\delta}_\square(G, G') = \min_{\tilde{G} \cong G} d_\square(\tilde{G}, G')$, where $\tilde{G}$ shares the same set of vertices with $G'$.

Next, we define the matrix infinity norm $\|.\|_\infty$ induced from the vector max norm.

**Definition A.3.** For a given matrix $A = (a_{ij})_{i=\overline{1,k}, j=\overline{1,n}}$, its infinity norm is computed as $\|A\|_\infty = \max_{1 \le i \le k} \sum_{j=1}^{n} |a_{ij}|$. In words, the matrix infinity norm is the max row sum.

**Proposition A.4.** *The matrix infinity norm is sub-multiplicative, i.e., $\|AB\|_\infty \le \|A\|_\infty \|B\|_\infty$.*

## B  PROOF OF THEOREM 3.2 AND MORE THEORETICAL RESULTS

In this section, we provide proofs for theoretical results in the main text. An illustration of theoretical results are given in Fig. 4.

*Proof of Theorem 3.2.* Denote $\varepsilon = \hat{\delta}_\square(G, G')$. Since $G \cong \mathcal{I}(G)$ and $G' \cong \mathcal{I}(G')$, it follows that $\hat{\delta}_\square(\mathcal{I}(G), \mathcal{I}(G')) = \hat{\delta}_\square(G, G') = \varepsilon$. From the definition of $\hat{\delta}_\square$ (Definition A.2), there exists $G^{\text{ind}} \in \mathcal{G}_k^{\text{ind}}$ so that $G^{\text{ind}} \cong \mathcal{I}(G)$ and $d_\square(G^{\text{ind}}, \mathcal{I}(G')) = \varepsilon$. Denote $\mathcal{E} = A_{G^{\text{ind}}} - A_{\mathcal{I}(G')}$, where $A_{G^{\text{ind}}}$ and $A_{\mathcal{I}(G')}$ are the adjacency matrices of $G^{\text{ind}}$ and $\mathcal{I}(G')$ respectively, it follows from the definition of $d_\square$ that $\|\mathcal{E}\|_\infty \le k^3 \varepsilon$ (since the absolute value of each element of $\mathcal{E}$ is less than $k^2 \varepsilon$ due to Definition A.1 of $d_\square$, and the infinity matrix norm is the max row sum where each row has $k$ elements). Finally, note that $A_{\mathcal{I}(G')} = A_{G'}$, we have

$$
\begin{aligned}
\|r_G - r_{G'}\|_\infty &= \|A_G V^{\text{ind}} - A_{G'} V^{\text{ind}}\|_\infty \\
&\le \|A_G - A_{G^{\text{ind}}} + \mathcal{E}\|_\infty \|V^{\text{ind}}\|_\infty \\
&= \|A_{\mathcal{I}(G)} - A_{G^{\text{ind}}} + \mathcal{E}\|_\infty \|V^{\text{ind}}\|_\infty \\
&\le \left(\|A_G\|_\infty + \|A_{G^{\text{ind}}}\|_\infty + \|\mathcal{E}\|_\infty\right) \|V^{\text{ind}}\|_\infty \\
&\le (2k + k^3 \varepsilon) \|V^{\text{ind}}\|_\infty \\
&= k(2 + k^2 \varepsilon) \|V^{\text{ind}}\|_\infty. \qquad \square
\end{aligned}
$$

Theorem 3.2 focuses on the distance between two single indirection representations. Now we move our attention to the distance between sets of indirection representations associated with isomorphic classes of graphs. For each $G \in \mathcal{G}_k$, denote $R_G = \{r_{\tilde{G}} : \tilde{G} \in \mathcal{G}_k, \tilde{G} \cong G\}$. According to

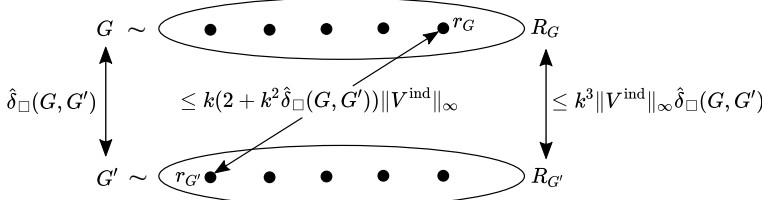

Figure 4: Illustration of Theorem 3.2 and Theorem B.1.

Theorem 3.2, diam $R_G \leq 2\|V^{\text{ind}}\|_\infty$ for every $G \in \mathcal{G}_k$, where diam stands for the diameter of a set. We now focus on the distance between $R_G$ and $R_{G'}$, and we use the popular Hausdorff metric to measure this distance. The distance between $R_G$ and $R_{G'}$ is meaningful in the sense that for any $r' \in R_{G'}$, we can find $r \in R_G$ so that $\|r - r'\|_\infty \leq d_H(R_G, R_{G'})$. If $d_H(R_G, R_{G'})$ is small and $r$ is observed, then $r'$ is likely to be treated similarly as $r$. The following last theorem shows that the Hausdorff distance $d_H$ with respect to the $\|.\|_\infty$ norm between $R_G$ and $R_{G'}$ also depends on the distance between $G$ and $G'$.

**Theorem B.1.** *Given two graphs $G \in \mathcal{G}_k$ and $G' \in \mathcal{G}_k$. The following inequality holds:*

$$d_H(R_G, R_{G'}) \leq k^3 \|V^{ind}\|_\infty \hat{\delta}(G, G'). \tag{4}$$

*Moreover, if $G$ and $G'$ are not isomorphic and rank $V^{ind} = k$, then $R_G \cap R_{G'} = \varnothing$.*

*Proof of Theorem B.1.* Denote $[G] = \{\tilde{G} \in \mathcal{G}_k : \tilde{G} \cong G\}$. We will prove that for every $G_1 \in [G]$, there exits $G_2 \in [G']$ so that $\|r_{G_1} - r_{G_2}\|_\infty \leq k^3 \|V^{\text{ind}}\|_\infty \hat{\delta}(G, G')$.

Following the definition of $\hat{\delta}_\square$, for $G_1 \in [G]$, there exists $G_2 \in [G']$ so that $\hat{\delta}_\square(G_1, G') = d_\square(G_1, G_2)$. On the other hand, since $G_1 \cong G$, it follows that $\hat{\delta}_\square(G_1, G') = \hat{\delta}_\square(G, G')$, and hence $d_\square(G_1, G_2) = \hat{\delta}_\square(G, G')$. This leads to

$$\begin{aligned}
\|r_{G_1} - r_{G_2}\|_\infty &= \|A_{G_1} V^{\text{ind}} - A_{G_2} V^{\text{ind}}\|_\infty \\
&\leq \|A_{G_1} - A_{G_2}\|_\infty \|V^{\text{ind}}\|_\infty \\
&\leq k^3 d_\square(G_1, G_2) \|V^{\text{ind}}\|_\infty \\
&= k^3 \hat{\delta}_\square(G, G') \|V^{\text{ind}}\|_\infty.
\end{aligned}$$

This means

$$d(r_{G_1}, R_{G'}) = \inf_{\tilde{G} \in [G']} \|r_{G_1} - r_{\tilde{G}}\| \leq k^3 \hat{\delta}_\square(G, G') \|V^{\text{ind}}\|_\infty$$

for every $G_1 \in [G]$. Similary, $d(r_{G_2}, R_G) = \inf_{\tilde{G} \in [G]} \|r_{G_2} - r_{\tilde{G}}\| \leq k^3 \hat{\delta}_\square(G, G') \|V^{\text{ind}}\|_\infty$ for every $G_2 \in [G']$. This leads to

$$d_H(R_G, R_{G'}) = \max \left\{ \sup_{G_1 \in [G]} d(r_{G_1}, R_{G'}), \sup_{G_2 \in [G']} d(r_{G_2}, R_G) \right\} \leq k^3 \|V^{\text{ind}}\|_\infty \hat{\delta}(G, G').$$

Finally, if rank $V^{\text{ind}} = k$ and $G$ and $G'$ are not isomorphic, suppose there exists $r \in R_G \cap R_{G'}$. Since $r \in R_G$, there exists $G_1 \in [G]$ so that $r = r_{G_1} = A_{G_1} V^{\text{ind}}$. Similarly, there exists $G_2 \in [G']$ so that $r = A_{G_2} V^{\text{ind}}$. This leads to $A_{G_1} V^{\text{ind}} = A_{G_2} V^{\text{ind}}$, and since rank $V^{\text{ind}} = k$, we obtain $A_{G_1} = A_{G_2}$. This means $G_1 \cong G_2$, and hence $G \cong G'$, which is a contradiction. Hence $R_G \cap R_{G'} = \varnothing$. $\square$

## C  WELL-DEFINEDNESS OF INDIRECTION REPRESENTATION

In this section, we consider the case when the canonical assumption (see Remark 2.1) is not obeyed. First, we need a more general definition for the indirection operator (Definition 2.2).

**Definition C.1.** Given an input sequence $X = (x_1, x_2, \ldots, x_k)$ and its corresponding graph $G_X \in \mathcal{G}_k$, the *indirection operator* $\mathcal{I}$ is a mapping from $\mathcal{G}_k$ to $\mathcal{G}_k^{\text{ind}}$ that maps $G_X$ to $\mathcal{I}(G_X)$ so that 1. $A_{G_X} = A_{\mathcal{I}(G_X)}$ and 2. if the $i$-th vertex of $G_X$ represents for $x_j$, then the $i$-th vertex of $\mathcal{I}(G_X)$ represents for $v_j^{\text{ind}}$.

Consider an input sequence $X = (x_1, x_2, \ldots, x_k)$ and its corresponding graph $G_X$ with adjacency matrix $A_{G_X}$, which is computed based on the assumption in Remark 2.1. The associated indirection representation computed by Eq. (1) is $r_X$, i.e., the $i$-th element of $r_X$ is the indirection representation for $x_i$.

Now consider an arbitrary graph $G'_X$ that also represents $X$ with adjacency matrix $A_{G'_X}$ and set of vertices $\{v'_1, v'_2, \ldots, v'_k\}$. This means there exists a permutation $\sigma$ (with an associated permutation matrix $P$) so that $v'_{\sigma(i)}$ represents for $x_i$ for all $i$, and $A_{G'_X} = P A_{G_X} P^\top$. Suppose that $G'_X$ is mapped to $\mathcal{I}(G'_X)$ with set of vertices $\{u'_1, u'_2, \ldots, u'_k\}$. By the definition of indirection operator (Definition (2.2)), $v'_i$ is mapped to $u'_i$ (so that $A_{G'_X} = A_{\mathcal{I}(G'_X)}$) and $u'_{\sigma(i)}$ represents $v_i^{\text{ind}}$ (since $v'_{\sigma(i)}$ represents for $x_i$). It follows that the associated indirection representation $r'_X$ with respect to $G'_X$ is computed as

$$r'_X = A_{G'_X} P V^{\text{ind}} = P A_{G_X} P^\top P V^{\text{ind}} = P A_{G_X} V^{\text{ind}} = P r_X.$$

This means the $\sigma(i)$-th element of $r'_X$ is the $i$-th element of $r_X$. On the other hand, after graph propagation, $u'_{\sigma(i)}$ represents for the $\sigma(i)$-th element of $r'_X$, which is the $i$-th element of $r_X$. Since $v'_{\sigma(i)}$ represents for $x_i$ and $v'_{\sigma(i)}$ is mapped to $u'_{\sigma(i)}$ by the indirection operator $\mathcal{I}$, it follows that the $i$-th element of $r_X$ is the indirection representation for $x_i$. This shows that the indirection representations of $x_i$'s are unchanged when the vertices of $G_X$ permute.

# D FULL EXPERIMENTAL RESULTS

## D.1 OOD IQ PROBLEMS

### D.1.1 FINE DATASET

We report full results with standard deviations of experiments on FINE dataset with trainset CIFAR10 and testset Omniglot in Table 5.

| Train set
Test set | CIFAR100
Omniglot | | | |
|---|---|---|---|---|
| | Trans. | Rot. | Shear | Scale |
| NTM | 25.5±1.7/**56.9±5.0** | 27.1±0.6/**31.4±1.7** | 29.8±1.3/**58.5±4.6** | 30.3±2.9/**61.0±3.9** |
| PrediNet | 26.7±0.9/**28.5±4.5** | 25.5±0.3/**28.3±0.5** | 25.6±0.4/**29.0±1.7** | 26.4±1.0/**29.9±1.7** |
| RelationNet | 25.4±0.5/**39.3±1.4** | 25.0±0.5/**34.0±0.4** | 25.1±0.2/**39.8±1.1** | 24.7±0.6/**50.2±1.0** |
| Transformer | 26.9±1.9/**63.3±3.7** | 27.2±1.1/**40.4±4.3** | 28.3±1.4/**65.3±2.8** | 30.6±3.0/**62.0±6.2** |

Table 5: Test accuracy (%) without/with InLay on FINE dataset with trainset CIFAR10 and testset Omniglot.

We report full results with standard deviations of experiments on FINE dataset with trainset MNIST and testset CIFAR10 in Table 6.

| Train set
Test set | MNIST
CIFAR100 | | | |
|---|---|---|---|---|
| | Trans. | Rot. | Shear | Scale |
| NTM | 35.2±1.6/**69.2±3.8** | 30.0±1.9/**34.5±5.6** | 29.6±1.1/**56.0±7.4** | 31.9±4.7/**48.7±7.8** |
| PrediNet | 27.5±1.4/**31.6±5.1** | 26.8±1.4/**27.8±1.5** | 26.4±1.4/**28.4±2.3** | 33.6±2.5/**35.9±6.7** |
| RelationNet | 25.6±0.5/**45.5±2.5** | 25.6±0.6/**35.2±1.2** | 25.0±0.5/**43.4±1.8** | 26.8±0.7/**44.8±4.1** |
| Transformer | 37.8±2.7/**63.9±6.5** | 32.0±1.5/**36.1±10.1** | 30.2±2.5/**53.5±6.8** | 29.2±3.6/**48.1±7.0** |

Table 6: Test accuracy (%) without/with InLay on FINE dataset.

### D.1.2 RAVEN DATASET

We report full results with standard deviations of experiments on RAVEN dataset in Table 7.

| Train configuration | Up-Down | 3x3Grid | Out-InGrid |
|---|---|---|---|
| Test configuration | Left-Right | 2x2Grid | Out-InCenter |
| LSTM | 28.8±3.1/**43.9±6.2** | 19.5±0.6/**25.1±2.4** | 42.1±1.3/**48.6±2.7** |
| Transformer | 15.2±0.3/**56.7±2.3** | 13.6±0.4/**26.0±0.9** | 16.6±4.5/**44.7±1.6** |
| RelationNet | 12.7±0.3/**58.5±2.0** | 12.4±0.4/**28.9±0.4** | 12.3±0.3/**51.7±2.8** |
| PrediNet | 13.7±0.3/**16.5±0.9** | 13.6±0.3/**14.3±0.6** | 14.2±0.6/**15.9±0.5** |

Table 7: Test accuracy (%) without/with InLay on RAVEN dataset.

## D.2 OOD Image Classification

We report full results with standard deviations of OOD image classification experiments in Table 8.

| Dataset | No data augmentation | | | With data augmentation | | |
|---|---|---|---|---|---|---|
| | Rot90 | Grayscale | Jitter | Rot90 | Grayscale | Jitter |
| SVHN | 11.8±0.4/**12.5±0.7** | 86.9±0.5/**91.8±0.7** | 87.2±0.2/**91.6±0.3** | 17.3±1.1/**19.1±1.6** | 92.9±0.2/**94.7±0.2** | 91.8±0.2/**93.4±0.2** |
| CIFAR10 | 25.7±0.1/**27.9±0.3** | 48.3±0.8/**62.7±0.2** | 46.6±0.9/**59.7±0.8** | 24.2±0.4/**27.3±1.3** | 54.2±0.4/**69.2±0.9** | 50.2±0.7/**68.4±0.5** |
| CIFAR100 | 13.4±0.4/**15.0±0.2** | 16.4±0.3/**23.9±0.4** | 18.1±0.2/**25.0±0.6** | 11.1±0.5/**14.0±0.4** | 18.5±0.7/**27.8±0.5** | 19.4±0.5/**28.7±0.5** |

Table 8: Test accuracy (%) without/with InLay on OOD classification task with ViT.

## D.3 Few-shot NLP Domain Adaptation

We report full results with standard deviations of few-shot NLP classification tasks on FewRel 2.0 dataset in Table 9.

| | ProtoNet | SNAIL | GNN | MTB |
|---|---|---|---|---|
| 5-way-1-shot | 64.7±1.5/**65.4±3.0** | 38.6±2.9/**60.6±4.9** | 36.7±2.9/**63.1±1.3** | 66.1±1.7/**68.9±1.0** |
| 10-way-1-shot | **49.1±2.0**/41.7±3.2 | 17.3±1.2/**34.9±5.8** | 27.1/**43.3±3.4** | 52.9±1.2/**54.1±1.3** |

Table 9: Validation accuracy (%) without/with InLay on FewRel 2.0.

## E 6-layer ViT vs. 7-layer ViT

We report results of 6-layer ViT and 7-layer ViT, along with 6-layer ViT equipped with InLay, in OOD CIFAR10 classification tasks in Table 10. Overall, average performance of 6-layer ViT and 7-layer ViT is not much different (38.2% and 38%, respectively), while 6-layer ViT equipped with InLay clearly improve performances with average accuracy of 43.1%.

## F Running time of InLay

We report running time (s/iter) of ViT and ViT+InLay on CIFAR10 dataset. Models are trained on a single Tesla V100-SXM2 GPU. Overall, ViT+InLay requires roughly 10% more computational time.

## G More ablation studies on tanh activation

Readers may observe in Fig. 3a that having no activation in InLay still achieves almost equal performance. However, that is just a special case; Table shows results of similar ablation experiments with NTM on the FINE dataset with different activation functions. Among all, the tanh activation achieves best average performance.

## H About input sequence length

### H.1 When the sequence is too long

When the number of nodes is large, graph neural networks usually suffer from the issue of over-smoothness, which is the phenomenon that all nodes become nearly the same after updated. However, we show that InLay may mildly suffer from this issue. In the OOD classification task, we increase

| | Image transformation | | Color transformation | | | | Average |
|---|---|---|---|---|---|---|---|
| | Rot90 | VFlip | Jitter | Invert | Blur | Grayscale | |
| 6-layer ViT | 25.7±0.1 | 32.1±0.6 | 23.2±1.1 | 39.4±2.8 | 60.2±0.5 | 48.3±0.8 | 38.2 |
| 7-layer ViT | 26.4 ±0.5 | 31.5±0.3 | 22.4±1.2 | 38.6±3.2 | **60.8 ±0.6** | 48.3 ±0.7 | 38 |
| 6-layer ViT + InLay | **27.9±0.3** | **35.0±0.9** | **24.9±0.7** | **48.3±3.3** | 59.5±1.2 | **62.7±0.2** | **43.1** |

Table 10: Test accuracy (%) without/with InLay on OOD CIFAR10 classification tasks with 6-layer ViT, 7-layer ViT, and 6-layer ViT equipped with InLay.

| | No data augmentation | With data augmentation |
|---|---|---|
| ViT | 0.095 | 0.100 |
| ViT+InLay | 0.108 | 0.110 |

Table 11: Running time (s/iter) of ViT and ViT+InLay on CIFAR10 dataset.

the number of nodes by duplicating each node 4 times, resulting in a graph with 196 nodes. The results are reported in Table 13. It can be observed that the test accuracy in the case of 196 nodes is not much different from the case of 49 nodes.

## H.2 WHEN THE SEQUENCE LENGTH IS NOT FIXED

All input sequences in our experiments are of fixed length. The case of varying input sequence length can be treated as fixed-length case if the maximum sequence length is known: we can use empty nodes to fulfill any sequence to reach that maximum length. We conduct experiments on OOD classification task to illustrate this idea: for each image patches sequence, we randomly remove some (2 to 6) patches, so that the resulting sequences have different lengths; we then use zero tensors for padding so that all sequences now have the same lengths. We train ViT+InLay on CIFAR10 dataset and test on images with grayscale distortion. The test accuracy is 61.4%, which is not much different from 62.7% of the fixed-length case.

A more challenging scenario is when the lengths of testing sequences are longer than training ones. Current design of InLay does not allow it to deal with this situation. We believe this is promising for future work.

## I COMPARISON WITH OTHER INDIRECTION APPROACHES

We compare InLay with different indirection approaches like ESBN (Webb et al., 2020b) and FINE (Pham et al., 2022) on FINE dataset. We incorporate InLay with Transformer as it shows the best performance among different models. For FINE, we use NICE backbone as suggested in the original paper. Results are shown in Table 14. Overall, Transformer+InLay shows competitive results with the best performances on 5/8 tasks and second-best performances on 2/8 tasks.

## J MORE ABLATION EXPERIMENTS ON CONTEXT NORMALIZATION

We further conduct ablation experiments to show the necessity of context normalization in InLay. Table 15 show performances of NTM and NTM+InLay, with and without context normalization. First, we may observe that InLay is really effective in the sense that NTM+InLay without context normalization is still far better than original NTM, both with and without context normalization. Second, context normalization helps to boost the performances of NTM+InLay by a large margin.

## K TRAINING DETAILS

### K.1 IQ OOD PROBLEMS

#### K.1.1 FINE DATASET

We use 3-layer $p4$-CNN encoder (Cohen and Welling, 2016) with kernel size 3 and padding 1 to encode raw $32 \times 32$ images to feature vectors of size 128. This encoder can help model adapt better

| Train set | MNIST | | | | |
| Test set | CIFAR100 | | | | |
| | Trans. | Rot. | Shear | Scale | *Avg.* |
| NTM+InLay(tanh) | **69.2** | **34.5** | 56.0 | **48.7** | **52.1** |
| NTM+InLay(none) | 58.1 | 31.2 | 52.5 | 47.8 | 47.4 |
| NTM+InLay(softmax) | 62.1 | 31.9 | **58.4** | 41.2 | 48.4 |
| NTM+InLay (relu) | 52.7 | 32.3 | 51.2 | 33.5 | 42.4 |
| NTM | 35.2 | 30.0 | 29.6 | 31.9 | 31.7 |

Table 12: Average test accuracy (%) of NTM (with or without InLay) on FINE dataset with different activation functions.

| | ViT | ViT+InLay (49 nodes) | ViT+InLay (196 nodes) |
| --- | --- | --- | --- |
| Test accuracy (%) | 48.3 | 62.7 | 62.5 |

Table 13: Experiment results of ViT+InLay with different number of graph nodes on OOD classification task on CIFAR10 dataset. Test distortion is grayscale.

with transformed images. We use Adam optimizer (Kingma and Ba, 2014) with learning rates ranging from $10^{-5}$ to $3 \cdot 10^{-4}$, depending on specific model and transformation. We train all models with batch size 32 in 200 epochs. The indirection representations are computed as in Eq. (1) with 1 attention head.

The training set contains 5,000 IQ problems, while testing set contains 10,000 IQ problems of unseen images and unseen rules. Specifically:

- With translation, models are trained on problems with translation vectors $(a, b)$ with $a \in \{0, 3, 6, 9\}$ and $b \in \{0, \pm 3, \pm 6, \pm 9\}$, and tested with $a \in \{-3, -6, -9\}$, i.e., train on problems with translations to the right and test on problems with translations to the left.

- With rotation, models are trained on problems with rotation angle $\alpha \in \{0°, 15°, 30°, \ldots, 180°\}$, tested with $\alpha \in \{195°, 210°, \ldots, 345°\}$.

- With shear, models are trained on problems with shear angles $(\alpha, \beta)$ with $\alpha \in \{0, 15°, 30°, 45°, 60°\}$ and $\beta \in \{0, \pm 15°, \pm 30°, \pm 45°, \pm 60°\}$, and tested with $\alpha \in \{-15°, -30°, -45°, -60°\}$.

- With scale, models are trained on problems with scale factor $\alpha \in \{1, 1.25\}$ and tested with $\alpha \in \{0.5, 0.75\}$, i.e., train with larger scale and test with smaller scale.

### K.1.2 RAVEN DATASET

We use 3-layer CNN encoder with kernel size 3 and stride 2 to encode $80 \times 80$ images to feature vectors of size 256. We use Adam optimizer with learning rates ranging from $10^{-4}$ to $3 \cdot 10^{-4}$ and gradient clipping 1. All models are trained with batch size 32 in 250 epochs. The indirection representations are computed as in Eq. (1) with 1 attention head.

We apply the Dynamic Residual Tree (DRT) as follows: we first apply DRT to feature vectors, then pass resulting vectors through InLay to obtain indirection representations, then apply DRT once again and these final resulting vectors will be the input for prediction models. Other details are similar to the original paper and codes are also adapted from the original paper.

### K.2 OOD IMAGE CLASSIFICATION

We use Adam optimizer with learning rate $5 \cdot 10^{-4}$. All models are trained with batch size 32 in 200 epochs. The indirection representations are computed as in Eq. (1) with 32 attention heads.

We use 6-layer ViT with patch size 8, 16 attention heads and dropout rate 0.1. The dimension of feedforward layer is 2048.

### K.3 FEW-SHOT NLP DOMAIN ADAPTATION

We use BERT to encode paragraphs to feature vectors of size 768. Models are trained with batch size 32 on 5-way-1-shot tasks and batch size 16 on 10-way-1-shot tasks. We use SGD optimizer with

| Train set | CIFAR100 | | | | MNIST | | | |
| Test set | Omniglot | | | | CIFAR100 | | | |
| | Trans. | Rot. | Shear | Scale | Trans. | Rot. | Shear | Scale |
| FINE | 27.3 | 37.6 | 18.7 | 19.5 | 47.7 | 45.2 | 48.8 | 26.6 |
| ESBN | 53.5 | 39.9 | 56.6 | 45.2 | 54.1 | **50.3** | **70.7** | **58.8** |
| Transformer+InLay | **63.3** | **40.4** | **65.3** | **62.0** | **63.9** | 36.1 | 53.5 | 48.1 |

Table 14: Average test accuracy (%) without/with InLay on FINE dataset.

| Train set | MNIST | | | | |
| Test set | CIFAR100 | | | | |
| | Trans. | Rot. | Shear | Scale | *Avg.* |
| NTM+InLay(with context norm.) | **69.2** | 34.5 | **56.0** | **48.7** | **52.1** |
| NTM+InLay (without context norm.) | 44.0 | **41.3** | 43.6 | 36.3 | 41.3 |
| NTM (with context norm.) | 35.2 | 30.0 | 29.6 | 31.9 | 31.7 |
| NTM (without context norm.) | 33.9 | 28.7 | 32.3 | 37.1 | 33 |

Table 15: Average test accuracy (%) of NTM (with or without InLay) on FINE dataset with different activation functions.

learning rate $2 \cdot 10^{-5}$. The indirection representations are computed as in Eq. (1) with 32 attention heads. Other details are similar to the original paper and codes are also adapted from the original paper.

