# OpenReview forum: "Improving Out-of-distribution Generalization with Indirection Representations"
_ICLR.cc/2023/Conference — ICLR 2023 poster_

### Official Review · Reviewer_JfzH · 2022-10-20

**Confidence:** 4
**Correctness:** 3
**Technical Novelty And Significance:** 3
**Empirical Novelty And Significance:** 3
**Recommendation:** 6

**Clarity, Quality, Novelty And Reproducibility:**

The paper is well-written and expresses its point clearly.

The authors first propose a general module that maps the input sequence into the graph-structured indirection representation, significantly improving the model generalization performance.

Although the module is a little similar to self-attention, InLay has the data-independent and trainable $V^{ind}$ that shows its novelty.

I think the module, InLay, proposed in this article is reproducible, but the author did not give the implementation code. I hope the authors can provide the codes in the future.

All in all, I think the outcomes and the writing are of high quality.


**Strength And Weaknesses:**

Strength:
1. The authors propose a generic module named Indirection Layer (Inlay), which can be plugged into different models to improve out-of-distribution generalization.
2. Tens of percent performance improvement results demonstrate the powerfulness of InLay.
3. The authors theoretically show the effectiveness of InLay which preserves the internal structures of graphs and the global structure of graph space.

Weaknesses:
1. The authors did not explain how to tackle the situations if the length of a sequence is not fixed.
2. The authors may want to do some ablation experiments to show that the improvement is caused by InLayer, not the context normalization that is used with InLayer.


**Summary Of The Paper:**

The authors propose a generic module named Indirection Layer (InLay), which can be plugged into different models to improve out-of-distribution generalization. InLay leverages the idea of indirection to redirect data representation based on a trainable set of symbols.
Specifically, InLay takes a sequence of objects as input and transforms the sequence into a new graph-structured indirection representation that leverages indirection and data internal relationships. The authors demonstrate that InLay is consistently effective in improving out-of-distribution generalization throughout a comprehensive suite of experiments.


**Summary Of The Review:**

The authors first propose a general module named Indirection Layer (InLay) for improving models’ generalization performance.
Specifically, the module maps the input sequences into the graph-structured indirection representations. The experiments show its powerfulness throughout some popular datasets.
Although the module is a little similar to self-attention, InLay has the data-independent and trainable $V^{ind}$ that shows its novelty.

---

> ### Author Response · Authors · 2022-11-18
> **Response to Reviewer JfzH**
>
> We thank the reviewer for their generous review. We would like to address the reviewer's concerns as follows.
>
> • Regarding the situations when the sequence length is not fixed: We thank the reviewer for the insightful comment. In the revision, we have discussed this point in Appendix H. In short, we conducted experiments on the OOD classification task in the non-fixed length case: for each image patches sequence, we randomly removed some (2 to 6) patches so that the resulting sequences have different lengths; we then used zero tensors for padding so that all sequences now have the same lengths. The experimental results show that InLay can adapt to this situation and the test accuracy is comparable to one with original setting.
>
> • Regarding ablation experiments with respect to context normalization: We thank the reviewer for the suggestion. In our experiments, we train all models (with and without InLay) under the same settings (with or without context normalization), which shows that the improvements are caused by InLay. In the revision, we added Appendix J to further illustrate this point on FINE dataset with NTM model: NTM+InLay without context normalization is still far better than original NTM with or without context normalization.
>
> Overall, in the revision, besides experimental results mentioned above, we have fixed some technical details regarding theoretical results as well as included Section 3.2 to discuss about the connection between InLay and structural analogy. In short, InLay is closely related to the idea of structural analogy, and theoretical results in Section 3.1 further draw a connection between InLay and cut distance, which is a useful metric to measure similarities between structures. We believe this may help to better position our InLay in the broader picture of generalization. We also provide the code for the OOD classification experiments for the purpose of reproducing.
>
> We once again thank the reviewer for their thoughtful comments. We hope that the reviewer will be satisfied with our responses, and will increase their score accordingly.

---

### Official Review · Reviewer_EM2y · 2022-10-20

**Confidence:** 4
**Correctness:** 3
**Technical Novelty And Significance:** 2
**Empirical Novelty And Significance:** 2
**Recommendation:** 6

**Clarity, Quality, Novelty And Reproducibility:**

The writeup of the paper is very clear. The hyperparameter details along with the empirical evidence presented suggest that the results are likely to be reproducible. The authors strongly emphasize that the proposed method is subtly different from self-attention and also demonstrate that it is empirically superior in appendix table 10. More results on this will help analyse the contributions of the work better.


**Strength And Weaknesses:**

1. The results across the tables in the main paper demonstrate that the proposed method indeed helps in generalization.
2. Figure 3 a) provides a satisfactory ablation of the proposed segments of InLay module. It is interesting to note that not having an activation layer to constrain the graph weights also performs almost equally well. Does this hold across other model-dataset pairs as well?
3. Leveraging a complete graph can be highly inefficient. I believe there should be experimental settings where the input sequence is large. In such cases, the proposed method could also struggle to learn good representations due to issues such as oversmoothness.
4. The computation of the adjacency matrix $a^{X}_{ij}$ in the first paragraph of page 3 doesn’t seem to be undirected.
5. In most of the experimental settings, using invariant networks and data augmentation also achieves the same goal. It will be highly relevant in the context of the work to provide comparison against some recent techniques while simultaneously reporting the runtime comparisons (one can see inducing the InLay layer may not increase the runtime as drastically compared to say - data augmentation). Similarly, the comparison to ESBN and FINE baselines on the IQ benchmarks seem necessary.
6. The standard deviation numbers provided in the appendix are fairly high across the board. It is evident that the InLay module makes the model predictions somewhat unstable. Do the authors have potential rectification/s for this?


**Summary Of The Paper:**

The authors propose a new module named “InLay” that can be plugged into different models across multiple modalities to improve out of distribution performance. The work leverages the idea of “indirection” to learn *indirection representations* for a given sequence of inputs. The rationale behind the proposed method is the key findings of Structure Mapping Theory that relationships between the objects are transferred during analogy rather than the attributes. More concretely, each data sample is viewed as a complete weighted graph where the weights are computed using a simple kernel between the pairs of objects in the input. The authors highlight the subtle difference of this scheme compared to the attention mechanism in the use of “tanh”. The input graph is then mapped to a new graph where the vertex embeddings are obtained from a trainable dictionary $V^{ind}$. Subsequently, a propagation scheme is applied to generate the indirection representations, as shown in eq 1. They also highlight the subtle difference that $V^{ind}$ is not computed based on the given input sequence and rather a standalone dictionary that represents the training data distribution as a whole. Furthermore, they provide an upper bound on the $l_{\infty}$ norm of the difference between indirectons representations based on a graph distance (the generalized rectangle distance) and $V^{ind}$. They also show that the indirection representation space is partitioned into disjoint subsets each of which correspond to a class of isomorphic graphs. Empirical evidence provided on different datasets across multiple modalities demonstrate the efficacy of the proposed method.


**Summary Of The Review:**

 Based on the questions, comments and concerns raised in the previous sections, I lean towards weak rejection of the work. I am willing to reconsider my score if the authors can answer the questions raised above.

---

> ### Author Response · Authors · 2022-11-18
> **Response to Reviewer EM2y**
>
> We thank the reviewer for their careful comments. We would like to address the reviewer's concerns as follows (the enumeration follows reviewer's comments).
>
> 2. We thank the reviewer for the thoughtful observation. Not having an activation layer may significantly decrease the performance of InLay, as we show in Appendix G in the revision.
>
> 3. We thank the reviewer for pointing out the potential problem of oversmoothness. However, we show in Appendix H that our method mildly suffers from oversmoothness when the number of graph nodes increases. Specifically, in the OOD classification task, we increase the number of nodes by duplicating each node 4 times, resulting in a graph with 196 nodes. The test accuracy in that setting is 62.5%, which is nearly the same with 62.7% in the original setting.
>
> 4. We thank the reviewer for the concern on undirectedness of graphs in InLay. In the revision, we have edited $\\mathcal{G}_k$ to be the space of directed weighted complete graphs (with no self-loop) with $k$ vertices and edge weights in $[-1,1]$. Other theoretical results are still maintained as the cut distance can be defined on directed weighted graphs.
>
> 5. We thank the reviewer for the recommendations. In the revision, we have reconducted the OOD classification experiments (Section 4.2) for two cases: with and without data augmentation. The results show that 1. ViT+InLay without data augmentation is still better than ViT with data augmentation in most cases; and 2. with the same training settings (with or without data augmentation), adding InLay into ViT is still beneficial. We included a running time comparison of ViT and ViT+InLay in Appendix F. Overall, ViT+InLay requires roughly 10% more computational time and achieves significantly better results as shown in the paper. We also included comparisons of Transformer+InLay with other indirection approaches (ESBN and FINE) on the FINE dataset in Appendix I. Overall, Transformer+InLay shows competitive results, achieving the best results on 5/8 tasks.
>
> 6. We thank the reviewer for their thoughtful observation. It should be noted that the problem of high standard deviations is likely to happen on IQ datasets (FINE and RAVEN). We believe two possible reasons are: 1. Performances of models without InLay are mostly nearly random (25% on FINE and 12.5% on RAVEN), so the models cannot learn anything other than randomly guessing. This keeps the standard deviations of those models low. 2. The FINE dataset generates new data each running time. Since we run models for 10 times to compute the means and standard deviations, their performances may largely vary depending on datasets they are trained on.
>
> Overall, in the revision, besides experimental results mentioned above, we have fixed some technical details regarding theoretical results as well as included Section 3.2 to discuss about the connection between InLay and structural analogy. In short, InLay is closely related to the idea of structural analogy, and theoretical results in Section 3.1 further draw a connection between InLay and cut distance, which is a useful metric to measure similarities between structures. We believe this may help to better position our InLay in the broader picture of generalization. We also provide the code for the OOD classification experiments for the purpose of reproducing.
>
> We once again thank the reviewer for their detailed comments. We hope that the reviewer will be satisfied with our responses, and will increase their score accordingly.

---

### Official Review · Reviewer_ttym · 2022-10-29

**Confidence:** 4
**Correctness:** 1
**Technical Novelty And Significance:** 3
**Empirical Novelty And Significance:** 3
**Recommendation:** 5

**Clarity, Quality, Novelty And Reproducibility:**

Overall, I found the paper somewhat unclear, with significant technical issues which would need to be addressed before this paper is ready for publication. The idea is somewhat novel, but due to some of the aforementioned technical issues I am not confident that one could reproduce the results from the paper alone, and I was unable to find a code reference.

**Strength And Weaknesses:**

### Strengths
1. The paper presents a large number of evaluations on a variety of tasks, the vast majority of which exhibit improved (often substantial) performance of their proposed method.
2. The authors perform an ablation study to analyze the impact of some (but not all, see point below) of the design decisions for InLay.
3. The related work section seems quite comprehensive, which is particularly useful for certain aspects (eg. Structure Mapping Theory) which most readers may be unfamiliar.

### Weaknesses
The most fundamental issue from a technical sense is that, to my understanding, there is an error in both proofs.
#### **Errors in Proofs**
1. The authors seem to rely on the idea that the max norm is sub-multiplicative, i.e. if $\lVert A \rVert = \max_{ij} |a_{ij}|$ then the authors make use of the (incorrect) inequality $\lVert AB \rVert \le \lVert A \rVert \lVert B \rVert$ in both proofs. I should note, it is not clear to me if this is actually the intended meaning of their max norm, since the statement of Theorem 3.2 says "$\lVert \cdot \rVert_\infty$ is the max norm on $\mathbb R^n$", whereas the elements which the authors calculate the norm of are (to my understanding) in $\mathbb R^{k \times n}$. If I have understood it correctly, however, the inequality mentioned earlier is incorrect, as can be observed by considering $A = B = \begin{bmatrix} 2 & 2 \\\\ 2 & 2\end{bmatrix}$.
2. In the proof of theorem 3.2, $f$ is not defined. I assume that it was intended that $f$ is the indirection operator, i.e. $f = \mathcal I$, however in that case there are lines which do not make sense (eg. the author's statement "From the definition of $\hat{\mathcal \delta}_\square$ there exists $G^\mathrm{ind} \in \mathcal G_k^\mathrm{ind}$ so that $G^\mathrm{ind} \cong f(G)$ and $d_\square(G^\mathrm{ind}, f(G'))=\varepsilon$." is unnecessary, as $\mathcal I(G) \in \mathcal G_k^\mathrm{ind}$ already.)

#### **General Theoretical Inconsistencies**
There were more general points of confusion throughout, most of which revolve around the theoretical formulation and motivation of the model.
1. $\mathcal G_k$ is defined as the space of all undirected weighted complete graphs $G$ with $k$ vertices, with edge weights belonging to $[-1,1]$, however the components of the adjacency matrix are clearly not symmetric. Perhaps it was intended that $\mathcal G_k$ is the space of directed weighted complete graphs. The problem gets more involved, however, as expressed in the following.
2. Throughout, the authors implicitly make use of a canonical indexing of vertices in a graph, but define things using a more standard representation of a graph (as a set of vertices and edges). This is most problematic in the definition of the indirection operator (Definition 2.1), which is ultimately not well-defined as currently described, which is as a map from $\mathcal G_k$ to $\mathcal G_k^\mathrm{ind}$ but which implicitly assumes an ordering of nodes. To highlight the issue, consider the graph $G$ with vertices $V = \\{5,8\\}$ and $E = \\{(5,8)\\}$. (Note: here the vertices are in $\mathbb R$, for simplicity in this example.) The definition of the indirection map requires an order on the vertices, but there is no inherent labeling - the vertices of a graph (both conventionally and as described in the paper) are a *set*. If we pick the labeling $x_1 = 5, x_2 = 8$ then the map takes $G$ to a graph with vertices $\\{v_1^\mathrm{ind}, v_2^\mathrm{ind}\\}$ and edge set $\\{(v_1, v_2)\\}$, however if we had labeled the vertices as $x_1 = 8, x_2 = 5$ then the image under $\mathcal I$ would have edge $\\{(v_2, v_1)\\}$. Therefore we have a contradiction, and thus $\mathcal I$ is not well-defined. There are two possibilities to remedy this situation. One is to instead consider not the set of graphs but rather the set of graphs with a given ordering, however this will add complexity and will become confusing as it is not classically how graphs are defined. The other is to reconsider whether this framework is really necessary. At the end of the day, the idea is simply that the output of InLay can be interpreted as having trainable value parameters which are decoupled from the data but combined using the pairwise similarity between input data. In my opinion, the existing framework seems unnecessarily complicated, and does not provide much insight. Either way, repairing this requires a substantial rewrite of sections 2 and 3.

#### **No Ablation of Fundamental Difference with Self-Attention**
The authors highlight the three points mentioned in my summary above as distinctions between InLay and self-attention, with a further emphasis that the fundamental distinction is point (3), namely that the values are not related to the input data. They do include an ablation of various components of InLay, but not of this crucial distinguishing factor. The authors also mention another distinguishing factor is that the output replaces the input data, as opposed to being added to it, and this would similarly be useful to assess with an ablation.


### Minor Issues / Typos / Suggestions
1. Section 3, first line, I believe "in the case of complete graphs" is unnecessary, this is true for all graphs.
2. Page 4, "preserves any topology on $\mathcal G_k$ induced from $\hat \delta_\square$" - this suggests that $\mathcal I$ is a homeomorphism, which is not the case (since it is not a bijection). Perhaps what was intended was "$\mathcal I$ is continuous with respect to any topology on $\mathcal G_k$ induced from $\hat \delta_\square$".
3. Page 4, and throughout: the authors emphasize the invariant properties of the indirection operator, however if one understands the definitions these properties are not particularly novel or surprising, since the properties are all defined in terms of the adjacency matrix and the indirection operator preserves the adjacency matrix. (This is another way in which I feel the current presentation results in unnecessary complexity.)
4. Page 4: "we can find $r \in R_G$" - isn't this the case that the inequality which follows holds for all $r\in R_G$?
5. Page 4: "it can be located close to some observed $r$" - I don't believe this means that $r'$ can be located close to some observed $r$, but rather: given $r'$, the distance from $r'$ to any observed $r$ is bounded by $d_H(R_G, R_{G'}) + 2 ||V^{ind}||_\infty$. (This bound of course depends on Theorem 3.2, whose proof currently has an error.)

**Summary Of The Paper:**

This paper proposes InLay, a new layer module which claims to capture internal relationships between input objects. Ultimately, for each input sequence $X \in \mathbb R^{k \times n}$ (interpreted as $k$ elements with $n$ features), the output of an InLay layer is given by
$\tanh\left(\frac{XQ(XK)^\mathrm T}{\sqrt{4n}}\right) V^\mathrm{ind},$
where $Q, K \in \mathbb R^{n \times 4n}$ are trainable weights and $V^\mathrm{ind} \in \mathbb R^{k\times n}$ are free parameters. The authors note the similarity of this operation to self-attention, with three mathematical distinctions: (1) $Q, K$ project into a higher-dimensional space, (2) $\operatorname{softmax}$ is replaced by $\tanh$, and (3) $V^\mathrm{ind}$ are free parameters, not a linear transformation of $X$. The authors motivate the design decisions by interpreting the inputs as vertices of a directed weighted graph whose edge weights are given by the $\tanh$ term in the layer, and thus the output of this layer can be viewed as propagating the (learned) features of $V^\mathrm{ind}$ along these weighted edges. The decoupling of $V^\mathrm{ind}$ from $X$ is intended to allow the layer to capture only internal relationships between objects, which is particularly beneficial for generalization in settings where the relationships between objects (eg. analogy, IQ test patterns) are the important factor. The authors evaluate on a large number of experiments of such tasks.

**Summary Of The Review:**

The idea is reasonable, however I felt it was not presented well. Some aspects of the motivation were more confusing than helpful, and the level of rigor was inconsistent, with some aspects (eg. emphasis on the indirection operator) being more complicated than necessary while other aspects (eg. proofs of theorems) had technical flaws.

---

> ### Author Response · Authors · 2022-11-18
> **Response to Reviewer ttym (1 of 3)**
>
> We thank the reviewer for their detailed and thoughtful comments. We would like to address the reviewer's concerns as follows.
>
> **About the errors in proofs**
>
> 1. About the inequality $\\|AB\\|_\\infty \\le \\|A\\|_\\infty\\|B\\|_\\infty$: We thank the reviewer for pointing this out. In the revision, we provided the definition for the $\\|.\\|_\\infty$ norm in Definition A.3, in which it is called the matrix infinity norm induced by vector max norm, and is computed as the maximum row sum. It follows in Proposition A.4 that $\\|.\\|_\\infty$ is sub-multiplicative (which is a special case of a more general result, which states that all matrix induced norms are sub-multiplicative). We also fixed other technical details in Theorem 3.2 so that the factor before $\\hat{\\delta}_\\square(G,G^\\prime)$ is $k(2+k^2\\varepsilon)$ instead of $2+k^2\\varepsilon$. However, we believe that this mistake does not affect the main idea of Theorem 3.2 and other claims based on this theorem: the distances between indirection representations are bounded by the corresponding cut distances $\\hat{\\delta}_\\square$, which shows that indirection representations are constrained by structural information of the input.
>
> 2. About the notation $f$ and the unnecessary statement in the proof of Theorem 3.2: We thank the reviewer for pointing out the notation mistake. In the revision, we have edited $f$ to be the indirection operator $\\mathcal{I}$. However, we believe that the statement “From the definition of $\\hat{\\delta}_\\square$, there exists $G^\\text{ind} \\in \\mathcal{G}_k^\\text{ind}$ so that $G^\\text{ind} \\cong \\mathcal{I}(G)$ and $d_\\square(G^\\text{ind}, \\mathcal{I}(G^\\prime))=\\varepsilon$” $(\\star)$ is necessary. What we have before this statement is $\\hat{\\delta}_\\square (\\mathcal{I}(G), \\mathcal{I}(G^\\prime)) = \\varepsilon$, which is equivalent to $\\displaystyle \\min_\\tilde{G} d_\\square(\\tilde{G}, \\mathcal{I}(G^\\prime))=\\varepsilon$ (over all $\\tilde{G} \\cong \\mathcal{I}(G)$), due to the definition of $\\hat{\\delta}_\\square$. From this, we can only conclude statement $(\\star)$ instead of $d_\\square(\\mathcal{I}(G), \\mathcal{I}(G^\\prime))=\\varepsilon$.
>
> **About theoretical inconsistencies**
>
> 1. About the symmetry of adjacency matrices: We thank the reviewer for the insightful comment. In the revision, we have edited $\\mathcal{G}_k$ to be the space of directed weighted complete graphs (with no self-loop) with $k$ vertices and edge weights in $[-1,1]$. Other theoretical results are still maintained as the cut distance can be defined on directed weighted graphs.
>
> 2. About the order of vertices in graph: We thank the reviewer for pointing this out with detailed examples. We agree that the set of vertices of graph is not indexed according to standard definition of graph, however we believe that the order of vertices is necessary when computing the adjacency matrix. In the revision, we have added Remark 2.1 and updated the definition of the indirection operator \inop (Definition 2.2 and the more general Definition C.1) to address this point. In short, we assume the canonical indexing assumption, which states that (when computing the adjacency matrix) the $i$-th vertex represents for the $i$-th element $x_i$ in the input sequence. The key points here are: 1. The input sequence is defined a priori (which is different to the reviewer's example, in which they assume $x_1=5, x_2=8$ and $x_1=8, x_2=5$ in two cases); and 2. The order of graph vertices when computing the adjacency matrix does not affect the indirection representations of $x_i$'s (see the proof in Appendix C in the revision; note that this does not mean the order of elements in the input sequence does not affect their indirection representations).

---

> > ### Author Response · Authors · 2022-11-18
> > **Response to Reviewer ttym (2 of 3)**
> >
> > Let us leverage the reviewer's example to further illustrate our point. As we are going to illustrate two cases, one is when the canonical indexing assumption is obeyed and one is when the canonical indexing assumption is not obeyed, we use the more general Definition C.1 for the indirection operator \inop in the following. Assume the input sequence is $(x_1,x_2)$. We are going to assign $x_1$ and $x_2$ to vertices of a graph G whose vertices are $p_1$ and $p_2$, and the image of $G$ through the indirection operator $\\mathcal{I}$ is $\\mathcal{I}(G)$ with vertices $q_1$ and $q_2$. There would be two cases:
> >
> > • $x_1$ is assigned to $p_1$ and $x_2$ is assigned to $p_2$ (the canonical indexing assumption is obeyed). The adjacency matrix is $A=\\left(\\begin{array}{cc}
> > 0 & a_{12}\\\\
> > a_{21} & 0
> > \\end{array}\\right)$. By definition, through the indirection operator, the adjacency matrix of $\\mathcal{I}(G)$ is also $A$ and $q_1$ represents $v_1^\\text{ind}$, $q_2$ represents $v_2^\\text{ind}$. The indirection representations are computed by $A\\left(\\begin{array}{c}
> > v_1^\\text{ind}\\\\
> > v_2^\\text{ind}
> > \\end{array}\\right)=\\left(\\begin{array}{c}
> > a_{12}v_2^\\text{ind}\\\\
> > a_{21}v_1^\\text{ind}
> > \\end{array}\\right)$, and thus the indirection representation for $x_1$ and $x_2$ is $a_{12}v_2^\\text{ind}$ and $a_{21}v_1^\\text{ind}$, respectively.
> >
> > • $x_1$ is assigned to $p_2$ and $x_2$ is assigned to $p_1$ (the canonical indexing assumption is _not_ obeyed). The adjacency matrix is $A=\\left(\\begin{array}{cc}
> > 0 & a_{21}\\\\
> > a_{12} & 0
> > \\end{array}\\right)$. By definition, through the indirection operator, the adjacency matrix of $\\mathcal{I}(G)$ is also $A$ and $q_1$ represents $v_2^\\text{ind}$, $q_2$ represents $v_1^\\text{ind}$. The indirection representations are computed by $A\\left(\\begin{array}{c}
> > v_2^\\text{ind}\\\\
> > v_1^\\text{ind}
> > \\end{array}\\right)=\\left(\\begin{array}{c}
> > a_{21}v_1^\\text{ind}\\\\
> > a_{12}v_2^\\text{ind}
> > \\end{array}\\right)$, and thus the indirection representation for $x_1$ and $x_2$ is $a_{12}v_2^\\text{ind}$ and $a_{21}v_1^\\text{ind}$, respectively (because the “second” vertex is representing for $x_1$ and the “first” vertex is representing for $x_2$).
> >
> > We can see the indirection representations for $x_1$ and $x_2$ in two cases are the same.
> >
> > **About ablation of fundamental difference with self-attention**: We thank the reviewer for the suggestion. In the revision, we included ablation studies for two additional cases: when the values are data-dependent and when the indirection representations are treated as relative positional encodings to be added into the input sequence. Overall, the performances in two cases are worse than the original settings of InLay, which confirms that the design choices of InLay are necessary.
> >
> > **About minor issues/typos**
> >
> > 1. Section 3, first line: In the revision, we have omitted the phrase "in the case of complete graphs".
> >
> > 2. and 3. We have omitted claims regarding topology invariants on $\mathcal{G}_k$ as we believe they does not strongly support our main claim. On the other hand, we believe invariants regarding graph spectrum, Laplacian and the cut distance $\\hat{\\delta}_\\square$ are important in our context: although they are obvious and somehow trivial (as mentioned by the reviewer), they show that InLay respects structural information of the input sequences, which is our main claim in the paper.
> >
> > 4 and 5. According to the definition of Hausdorff distance (see, e.g., [Wikipedia](https://en.wikipedia.org/wiki/Hausdorff_distance)), for a given $r^\\prime \\in R_{G^\\prime}$, we have $d(R_G, r^\\prime) \\le d_H(R_G,R_{G^\\prime})$ where $d(R_G, r^\\prime)=\\displaystyle \\inf_{r \\in R_G} \\|r-r^\\prime\\|_\\infty$. This means we can only conclude there exists $r \\in R_G$ such that $\\|r-r^\\prime \\|_\\infty \\le d_H(R_G, R_G')$ (by $R_G'$ we mean $R$ of $G^\\prime$, for some reason MathJax on OpenReview cannot display it here). If the Hausdorff distance between $R_G$ and $R_G'$ is small and $r$ is observed, then $r^\\prime$ is likely to be treated similarly as $r$. We have revised our manuscript to avoid vaguity.

---

> > > ### Author Response · Authors · 2022-11-18
> > > **Response to Reviewer ttym (3 of 3)**
> > >
> > > After all, we would like to defend our position that the method is not complicated and does provide new insights:
> > >
> > > • Our method is not complicated:
> > >
> > > – Through a comprehensive suite of experiments, we have shown that InLay can be easily incorporated into different deep learning models (Transformer, NTM, RelationNet, ProtoNet, GNN, etc.) with different encoder architectures (p4-CNN, BERT, or no encoder in the case of ViT). This shows great simplicity and flexibility of InLay as a generic plug-in module.
> > >
> > > – Although the theoretical results may look complicated, the implementation of InLay is simple with around 10 lines of code for the main module and some other minor modifications (e.g., the tanh activation instead of softmax) in the attention head. We included the code of OOD classification experiments in the Supplementary for the purpose of reproducing.
> > >
> > > • Our method does provide new insights:
> > >
> > > – The effectiveness of InLay shows that the inductive bias of indirection can be injected into deep learning models by a simple layer instead of designing specific models.
> > >
> > > – Design choices and theoretical results of InLay suggest a close connection between InLay and structural analogy (see Section 3.2 in the revision), which is believed to be necessary for extreme generalization. The connection drawn in Section 3.2 also helps to position InLay in the broader picture of generalization.
> > >
> > > – Overall, InLay proposes a generic method to abstract concrete data, which is important as human's thinking is believed to be abstract and symbolic.
> > >
> > > Once again, we would like to thank the reviewer for their careful and detailed comments. We hope that our response has addressed the reviewer's concerns, and that the reviewer will increase their score accordingly.

---

> > > > ### Comment · Reviewer_ttym · 2022-12-02
> > > > **Thank you, only minor technical concerns remain**
> > > >
> > > > Thank you for the comprehensive rebuttal, and my apologies for not responding sooner.
> > > >
> > > > I have read through the author's response as well as the new version of the paper. The most important aspect for me was the correctness of their proof, and after their updates I now agree with their current version.
> > > >
> > > > (Note: there is a minor typo on page 14, it should state $\operatorname{diam} R_G \le 2k\lVert V^\text{inf} \rVert_\infty$.)
> > > >
> > > > The other aspect which I highlighted originally was the dependence on order of nodes. This is a very subtle technical issue, but an important one as it is intimately related to the fundamental claims and focus of the work. The authors have added a remark and an expanded section in the appendix discussing the canonical indexing assumption, however this does not actually address the flaw, which is that the function $\mathcal I$ is still not well-defined as currently written - it may result in different outputs for elements which are considered to be the same in the input space, because (crucially) the input space is $\mathcal G_k$. The authors addressed the canonicalization at the level of the adjacency matrix, but this is not where the problem was.
> > > >
> > > > I did read and appreciate the author's detailed comment and example as well, however this is actually pointing out that the map taking $A_G$ to $r_G$ is permutation equivariant with respect to the ordering of the nodes. That's true, but that is not quite the issue I was raising here.
> > > >
> > > > The problem actually presents itself even earlier-on than $\mathcal I$. $\mathcal G^{ind}$ is defined as the set of graphs with $k$ vertices, $\{v_1, \ldots, v_k\}$, where each $v_i \in \mathbb R^n$ for some $n$. This is already an issue, as if $v_i = v_j$ then it actually has fewer than $k$ vertices. A nice solution which would avoid both issues is to consider functions on indices, i.e. $f: [k] \to \mathbb R^n$. Then the graphs themselves are not changing, rather the induction function simply swaps this function for another one.
> > > >
> > > > At the end of the day, it is a rather subtle and nuanced problem, but it is important (after all) to get these technical details correct.

---

> > > > > ### Author Response · Authors · 2022-12-10
> > > > > **Response to Reviewer ttym**
> > > > >
> > > > > We thank the reviewer for their generosity, and we would like to address their remaining concerns as follows.
> > > > >
> > > > > * About the typo on page 14: We thank the reviewer for pointing out our typo on page 14. Unfortunately we can not update our revision now; we will fix that typo in the camera-ready version if we have a chance to do so.
> > > > >
> > > > > * About the concern on the order of nodes: We agree with the reviewer that this is a subtle technical detail, however we also believe that it is necessary to get it right. We thank the reviewer for their recommendation on using functions on indices to avoid ambiguity in the case $v_i = v_j$ for some $i\\neq j$. This recommendation is elegant and reasonable, however we believe that the notation of the *sequence* $v_1, v_2, \\ldots, v_k$ may already indicate the same thing. The reviewer can take a look at Definition 2.7 in Rudin's classic book [1] for a formal definition of a sequence, in which a sequence is defined as a function defined on the set of positive integers (this definition is for the general case of infinite sequences; finite sequences can be defined in a similar way with index set as recommended by the reviewer). Rudin also notes that the terms of a sequence need not be distinct. For the case of our InLay, we believe that $v_i$'s need not be distinct either: for example, two images at two different positions in an IQ test can be identical (e.g., see [2], the second image in Row 1 and the second image in Row 3 are identical), however we can not omit any of them. We would like to adopt the elegant term "permutation equivariant" in the reviewer's comment along with the above explanations to improve our manuscript in the camera-ready version (if we have a chance to do so).
> > > > >
> > > > > We once again thank the reviewer for their generosity and deep understanding of our work. We hope that the reviewer is satisfied with our response, and the reviewer may further increase their score accordingly.
> > > > >
> > > > > [1] [Rudin, Walter. Principles of mathematical analysis. Vol. 3. New York: McGraw-hill, 1976.](https://web.math.ucsb.edu/~agboola/teaching/2021/winter/122A/rudin.pdf)
> > > > >
> > > > > [2] [An IQ test with identical images at different positions - See Problem 2 of Brain Matrix in this link](https://rigorousthemes.com/blog/best-free-iq-tests-online/)

---

> > > > > > ### Comment · Reviewer_ttym · 2022-12-14
> > > > > > **Further clarification re: graph node ordering**
> > > > > >
> > > > > > Thanks so much for your continued attention to this issue.
> > > > > >
> > > > > > I also agree that a sequence is equivalent to a function which is defined on the indices. Conventionally, and indeed in your current paper, however, Graphs are not defined as a tuple of *sets*, $G = (\mathcal V, \mathcal E)$. The set $\mathcal V$ is still a *set*, even if the elements are indexed, i.e. $\mathcal V = \{v_1, \ldots, v_n\}$. In fact, assigning the elements of this set indices is, itself, not a well-defined operation for exactly this reason, however it can still be useful if the resulting operations performed are permutation invariant. For example, saying "Let $S$ be any set of three elements, $S=\{s_1, s_2, s_3\}$, and let $f(S) = s_1$", this is not well-defined. On the other hand, if we say "Let $S$ be any set of three elements, $S=\{s_1, s_2, s_3\}$, and let $f(S) = s_1+s_2+s_3$, then this is fine - we were just using the indices to enable us to write down the definition of a function, but it is, in fact, a function, because for any two sets with the same elements the output of the function is the same. This fact is actually quite central to graphs, as it encodes the fact that the order of the nodes is *not* part of what makes two graphs different. This aspect of graphs is also quite fundamental to your paper, which is why I emphasize it so strongly.
> > > > > >
> > > > > > If you wanted to switch to using a sequence for graphs that could potentially be fine, but there are a number of potential pitfalls to be aware of. First and foremost is that this is not how graphs are traditionally defined, and since this is such a subtle point I think the distinction here may raise more confusion with people who conventionally are used to immediately jumping to the adjacency matrix of a graph which, after all, assumes an ordering of the nodes. Secondly, as mentioned above, part of the very premise of InLay is to design an architecture which is invariant to the ordering of nodes, and seems almost antithetical to start off by switching to graphs on a sequence of nodes. Third is that taking this unconventional approach means double-checking all other areas where you may have implicitly relied upon the standard definition of a graph which does not provide an order to the vertices. These may be subtle and hard to uncover, and so I would recommend not proceeding this way.
> > > > > >
> > > > > > I think the "function on the nodes" approach I suggested in my previous reply is the cleanest way to formalize this, as it should allow you to nicely decouple the graph structure from the node representation. In any event, I appreciate the time and attention the authors have given to this point, and I'm sure that with additional deliberation they will come up with a solution for the camera ready.

---

### Official Review · Reviewer_r9MA · 2022-11-03

**Confidence:** 3
**Correctness:** 4
**Technical Novelty And Significance:** 2
**Empirical Novelty And Significance:** 3
**Recommendation:** 8

**Clarity, Quality, Novelty And Reproducibility:**

Clear writing. Assuming the implementation will be open, other folks could find the layer useful.

**Strength And Weaknesses:**

Strength: The proposed method is easy to apply as it can be naturally added on top of representations by other base models. In addition, it shows nice boosts in appropriate settings and for OOD image classification.

Weakness: Current experiments mostly involve applications to simple images in synthetic settings (like IQ tests), or synthetic transformations to images like rotations. It is not clear at this point whether InLay can capture implicit and complex relations between natural images. In addition, all in all the layer is very similar to a self-attention layer.

**Summary Of The Paper:**

The authors present an layer in a neural network called Indirection Layer (InLay). InLay takes
a sequence of objects as input and transforms the sequence into a new indirect graph-structured
representation. In this graph vertices are the objects, and edges are similarity scores in the range [-1,1] that are learned. The similarity weights are then used to modify a representation V_ind of the vertices which is not computed based on the data X (it is a different set of vectors). This representation, that is contextualized by other members of the sequence is than used for prediction. All in all this is very similar to self attention, however the attention scores are in [-1,1] and V_ind is not computed based on the data X.
The goal of InLay is that graph representations of different two sequences that have similar internal structure should be similar, and thus help in OOD generalization.

InLay can be used as an additional layer on top of different architectures, e.g., Transformers or LSTMs.
Experiments of InLay are done for the datasets FINE and RAVEN, where a sequence of images with an internal relation are shown and the model should predict which image should complete the sequence. For both datasets, substetianl improvement is achieved by using InLay. In addition, the method is applied to OOD image classification tasks and shows nice gains.

**Summary Of The Review:**

InLay is a neural layer for representing objects in a sequence that share relations between them (e.g., rotation). InLay seems like a good direction towards capturing relations between objects that are useful for different classification tasks. The layer could be added in a simple manner on top of different architectures. Currently, the method is applies to synthetic tasks, and it is left for future work to find out whether the learned representation is useful in realistic settings.

---

> ### Author Response · Authors · 2022-11-18
> **Response to Reviewer r9MA**
>
> We thank the reviewer for their generosity. We would like to address the reviewer's concerns as follows.
>
> • “It is not clear at this point whether InLay can capture implicit and complex relations between natural images”: We thank the reviewer for pointing this out. Our experiments have been designed with increasing difficulties, including real images from CIFAR10&100 datasets, to clarify the effectiveness of our InLay. We start with synthetic IQ datasets (FINE and RAVEN) to have better control over OOD testing circumstances (e.g., use images from different datasets when testing), then we further conduct OOD classification experiments on real-image CIFAR10&100 datasets. Finally, we test InLay on few-shot NLP tasks with texts from real corpora (Wikipedia and PubMed). We believe that our experiments are sufficiently comprehensive to prove the effectiveness of InLay on both synthetic and real datasets in different domains (images and texts).
>
> • “The layer is very similar to a self-attention layer”: As mentioned in Section 2, the key difference between InLay and a self-attention layer is the values in InLay are trainable and data-independent, while the values in self-attention are data-dependent. In the revision, we included an ablation study (see Figure 3a) to show that trainable values in InLay are more effective than data-dependent ones in self-attention layer.
>
> In the revision, we additionally discuss the connection between InLay and structural analogy (see Section 3.2), which is an important concept regarding generalization. Such connection can help to position InLay (along with theoretical results in Section 3.1) in the broader picture of OOD generalization. We also provide the code of the OOD classification experiments for the purpose of reproducing.
>
> We once again thank the reviewer and hope that the reviewer will be satisfied with our responses.

---

### Decision · Program_Chairs · 2023-01-20

**Decision:**

Accept: poster

**Justification For Why Not Higher Score:**

Mostly due to the absence of more downstream evaluations. Overall scores are positive but not excited enough to merit a spotlight or oral.

**Justification For Why Not Lower Score:**

The paper should be accepted as 3/4 voted for acceptance and the last reviewer mostly wanted to encourage authors to hash out some final technical details in the theoretical formulation.

**Metareview: Summary, Strengths And Weaknesses:**

The authors propose a new layer that improves generalization to out-of-distribution samples.
Strengths: clear work, nice improvements across many experiments
Weaknesses: Evaluation on real end tasks and some disagreement about the theoretical formulation.


**Note From Pc:**

if the above contains the word "oral" or "spotlight" please see: "oral" presentation means -> notable-top-5% and "spotlight" means -> notable-top-25%. As stated in our emails, we are disassociating presentation type from AC recommendations

**Summary Of Ac-Reviewer Meeting:**

Discussion on whether the corrections that the authors have made make the proofs now correct.
Reviewers agreed that almost all theoretical questions have been answered.